# Group-wise Data Ordering: Enhancing Instruction Tuning of Large Language Models via Embedding Proximity

Yiwen Ye [1,2]  Boyuan Jiang [1]  Xiaobin Hu [3]  Shengzhi Wang [1]  Xiaozhong Ji [1]  Jinghao Lin [1]  Deli Yu [1]  Jiale Chen [1]  Kai Wu [1]  Haihua Yang [1]  Yong Xia [2]

## Abstract

Instruction tuning (IT) is a central mechanism for aligning large language models (LLMs) with user intent. In practice, randomly shuffling the training set is a simple yet surprisingly strong baseline. However, it overlooks latent structure, such as domain and reasoning depth, and thus interleaves heterogeneous objectives, which can induce gradient conflicts and diminish effective optimization progress. To this end, we propose **EP-Order**, an embedding-proximity-based data-ordering paradigm for IT of LLMs. Unlike previous paradigms that derive order from **per-example** scores, EP-Order explicitly accounts for **inter-sample** correlations by operating in representation space. EP-Order trains a warm-up model on a small subset of data (*e.g.*, 10%), embeds all training samples for clustering, and ranks these clusters according to embedding proximity. To handle sharp gradient changes at cluster transitions and alleviate catastrophic forgetting under cluster-based training, we introduce mixed regions that interleave samples from the previous, current, and next clusters, thereby stabilizing learning. Extensive experiments on 14 benchmarks spanning vision-language, text-only, and hybrid thinking/no-thinking scenarios show that EP-Order achieves broadly consistent improvements over random shuffling. These results suggest that exploiting embedding-level data structure offers an effective and general direction for improving IT on complex, high-conflict training data.

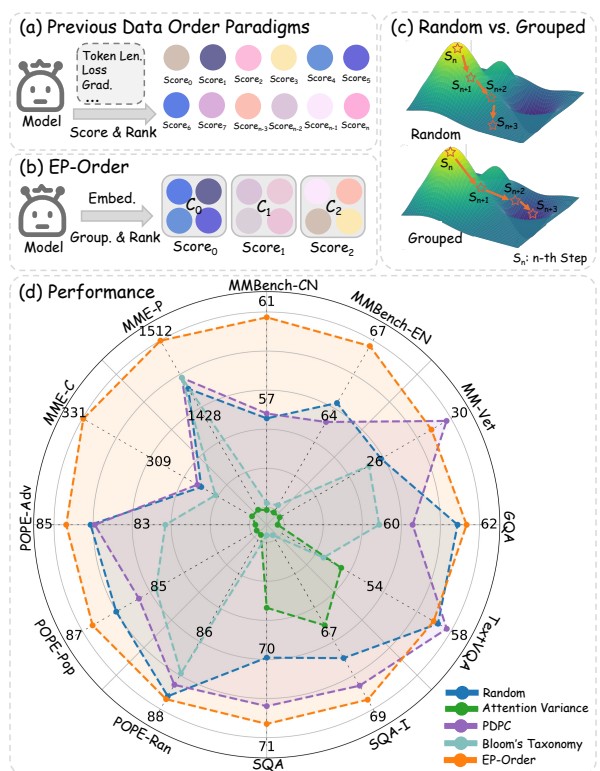

*Figure 1.* (a) Prior work assigns per-sample difficulty scores (*e.g.*, token length and loss) and orders training accordingly. (b) Our EP-Order schedules training at the group level: we cluster examples by embedding proximity, assign group-level scores, and train by groups rather than individual samples. (c) Illustration of the optimized paths of the Random paradigm (with smaller ER) and the Grouped paradigm (with larger ER) under the same number of steps. The Grouped paradigm moves faster and more accurately towards the optimization goal. (d) Generalization performance of EP-Order and four competing paradigms on 7 benchmarks.

## 1. Introduction

Instruction tuning (IT) has become a central step for aligning large language models (LLMs) with user intent, bridging unsupervised/supervised pretraining and post-training alignment. By fine-tuning on instruction–response pairs, LLMs acquire stronger task generalization and safer, more helpful behavior, making them more suitable for providing satisfac-

[1]Bytedance, China [2]Department of Radiology, Ningbo No. 2 Hospital, Ningbo, China [3]National University of Singapore, Singapore. Correspondence to: Haihua Yang <yanghaihua@bytedance.com>, Yong Xia <aleckxia@gmail.com>.

tory results.

While LLMs receive most of the attention, data is the primary driver of IT effectiveness with the guidance of the scaling law. Previous work mainly focuses on efficient IT, *i.e.*, data selection, showing that carefully designed mixtures and small, high-quality supervision can rival or exceed training with much larger but noisier corpora. Representative strategies include self-guided difficulty scoring after a brief warm-up (Li et al., 2024), gradient-based valuation (Liu et al., 2025), training-free intrinsic criteria (Bi et al., 2025), and adaptive online selection (Lee et al., 2025). These methods largely optimize what to keep (and how much), rather than how to organize the data.

A complementary and increasingly important perspective is to optimize how we arrange the training data we already have. Most prior work (Liu et al., 2024a; 2023; Zheng et al., 2024) follows a Random shuffling paradigm for training data, which is simple and often robust. However, random shuffling indiscriminately interleaves heterogeneous samples during optimization, implicitly ignoring latent structure such as difficulty, domain, modality, or reasoning depth. A prior study on multi-task optimization has shown that such unstructured interleaving of heterogeneous objectives can induce gradient conflicts and reduce effective optimization progress (Yu et al., 2020). This suggests that random mixing, while convenient, may be suboptimal in a high-conflict training paradigm. A straightforward approach is curriculum learning (CL), which orders examples from 'easy' to 'hard' using length-, attention-, entropy-, or loss-based heuristics (Kim & Lee, 2024; Jia et al., 2026). Beyond such proxies, model-aware evaluation strategies, *e.g.*, answer-token entropy (Goncharov et al., 2026), perplexity difference (Zhang et al., 2025b;a), and learnability–quality scoring (Dai et al., 2025), explicitly account for the dynamics of learning. However, these strategies derive per-example difficulty and largely ignore inter-sample interactions (see Fig. 1(a)), which may be a more critical bottleneck for building stronger IT models, as our experiments in Section 4.2 show that recent CL-based paradigms fail to outperform the Random paradigm.

To verify this intuitive insight, we experimented with evaluating the metrics of efficiency ratio (ER)[1], that larger ER yields larger effective progress per step (theoretical analysis is introduced in Section 3.2), for orders obtained from the Random paradigm or the group-based order. Specifically, for the Random paradigm, we randomly extracted 15 sets of samples from the LLaVA fine-tuning dataset (Liu et al., 2024a), and each set contains 128 samples (a mini-batch). For the group-based paradigm, we obtained groups based on embedding clustering and randomly extracted sets from

---

[1]ER is the norm of the weighted mean of per-sample gradient directions within a mini-batch.

each cluster, each also contains 128 samples. We calculated the ER for each set and regarded the average value for each paradigm. The result shows that the ER of the group-based order (0.71) is significantly larger than the random-based one (0.54), suggesting that the group-based order is able to provide more sufficient information utilization by reducing gradient cancellation. We illustrated the value of larger effective progress, as shown in Fig. 1(c), which moves faster and more accurately towards the optimization goal.

Motivated by this, we ask: *Can IT be improved by reorganizing data so that semantically similar examples are learned together, with CL operating over groups rather than isolated items?* Our core insight is that embedding-level proximity serves as a practical proxy for mutual influence during optimization: samples that are nearby in representation space tend to induce compatible gradients and are thus natural to co-train. Therefore, we propose **EP-Order**, an **E**mbedding-**P**roximity-based paradigm that produces a group-based data **Order** for LLM's IT, as shown in Fig. 1(b). EP-Order first trains a warm-up model on a small subset of data (*e.g.*, 10%), then uses the model to embed all training examples and clusters them in the resulting representation space. We construct a curriculum over clusters and train with overlapping 'mixed regions' that interleave examples from the previous, current, and next clusters to stabilize learning and mitigate catastrophic forgetting. We evaluate EP-Order across three application settings: vision–language, text-only, and hybrid thinking/no-thinking, covering 14 benchmarks (22 sub-benchmarks). While experiments often find random shuffling to be a strong baseline, EP-Order surpasses it on most benchmarks shown in Fig. 1(d).

The main contributions of this paper are as follows:

- We identify group-wise data ordering as an underexplored lever for IT and conduct a theoretical analysis of the advantage for group-based data ordering. We, therefore, introduce EP-Order, which organizes training by feature-proximity clusters rather than per-example difficulty.

- We instantiate EP-Order via (i) a warm-up stage to obtain representations, (ii) clustering to form semantically coherent groups, and (iii) an overlapping mixed-region schedule that maintains stability and reduces forgetting.

- We conduct comprehensive experiments on 14 benchmarks spanning vision–language, text-only, and hybrid thinking/no-thinking text-only scenarios, showing that EP-Order achieves superior generalization compared with random shuffling and competing ordering paradigms.

## 2. Related Work

The effectiveness of IT for large models (LMs), *e.g.*, multi-modal LLMs and LLMs, depends not only on data quality but also on how the training data is organized and presented during training. Our work builds on two lines of research: curriculum learning strategies for LMs and data optimization competition in multi-domain/multimodal data.

### 2.1. Curriculum Learning Strategies for LMs

**Static difficulty heuristics.** A straightforward way to impose a curriculum is to sort supervision examples from 'easy' to 'hard' using proxies defined on inputs, model traces, or early training signals. Kim and Lee (Kim & Lee, 2024) systematically compare length-, attention-, and loss-based difficulty measures under various training epochs, reporting small and inconsistent gains over random shuffling. EntropyLong (Jia et al., 2026) identifies high-entropy positions in documents, retrieves semantically relevant contexts from large corpora, and evaluates utility by checking whether prediction entropy is reduced. **Model-aware difficulty and preference.** Rather than static heuristics, model-centric scores measure what the current or a reference model finds hard. Complexity-aware fine-tuning (Goncharov et al., 2026) uses answer-token entropy computed by the student to split data, applying standard IT to the low-entropy split and distilled CoT to the high-entropy split. PDPC (Zhang et al., 2025b) defines a Preference Curriculum by measuring a sample's perplexity difference (PD) between a weak and a strong model, scheduling high-PD data later with an S-shaped pacing function. FRAME (Zhang et al., 2025a) partitions pre-training data by two axes, perplexity (PPL) and PD, and orders stages from high- to low-PPL and from low- to high-PD. DELT (Dai et al., 2025) frames data efficacy as three levers, scoring → selection → ordering, introducing Learnability–Quality Scoring, and Folding Ordering to counter forgetting and distribution bias, elevating ordering to a first-class design choice alongside filtering. **Scheduling beyond strict 'easy-to-hard'.** Human-inspired and competence-aware curricula relax strict monotonic schedules. Lee et al. (Lee et al., 2024) synthesize and stage instructions to emulate human learning sequences, while CAMPUS (Li et al., 2025) adapts difficulty to model progress. Zhang et al. (Zhang et al., 2026) systematize LLM curricula by three scheduler families (strict, pacing, interleaved) and six model-agnostic difficulty signals (compression ratio, lexical diversity, readability, token length, fertility, perplexity). They conclude that ordering itself matters, curricula, even as brief warm-ups, accelerate learning, and that information-density/lexical-richness signals are more reliable than raw perplexity. Despite their differences, the above strategies primarily derive order from **per-example** signals. In contrast, we advocate moving beyond isolated items to explicitly account for **inter-sample** correlations, and we introduce a group-wise ordering framework for IT.

### 2.2. Data Optimization Competition in Multi-domain/Multimodal Data

Data optimization competition refers to interference among heterogeneous corpora (domains/modalities) when jointly fine-tuning a single model, whereby only part of the data 'wins' optimization and the rest is underfit. Both theory and practice document this phenomenon: multi-modal/multi-domain joint training can suffer from modality competition and mismatched learning speeds (Huang et al., 2022). Gradient balancing (*e.g.*, Gradient Blending) reweights modality/task contributions to reduce interference and avoid overfitting to the easiest signal (Wang et al., 2020). Canonical gradient-geometry methods cast multi-task learning as multi-objective optimization (MGDA) to find a Pareto-descent direction (Sener & Koltun, 2018), perform gradient surgery when conflicts occur (PCGrad) (Yu et al., 2020), regularize toward the worst local improvement while optimizing the average loss (CAGrad) (Liu et al., 2021), or normalize magnitudes to equalize training rates (GradNorm) (Chen et al., 2018), thereby mitigating dominance. More recent schemes modulate both gradient magnitudes and directions to directly counter modality imbalance. In sequential IT, Continual LLaVA (Cao et al., 2024) frames continual instruction tuning as parameter-efficient embedding adaptation, freezing the backbone and introducing dual increment embeddings, to achieve rehearsal-free continual learning and reduce optimization competition across domains, capabilities, and datasets. Prior work primarily measures conflict (*e.g.*, gradient dot products/influence) and manages it via mixture scheduling or gradient reweighting. We found that embedding proximity can result in high ER within a mini-batch, therefore, providing more sufficient information utilization and effective learning.

## 3. Approach

### 3.1. Problem Definition

Let the (multimodal) IT dataset be $D_{it} = \{(x_1, y_1), \ldots, (x_t, y_t)\}$ with $t$ samples. The prevalent practice is random shuffling: $D_{it}$ is uniformly shuffled prior to training. While simple, this ignores informative structure and potential conflicts among samples. Our goal is to design an efficient ordering paradigm, EP-Order, that improves optimization and generalization with minimal overhead.

### 3.2. Group-Order vs. Random-Order Learning

Consider a mini-batch of size $B$. Let the per-sample gradient be

$$g_i \in \mathbb{R}^D, i = 1, ..., B. \tag{1}$$

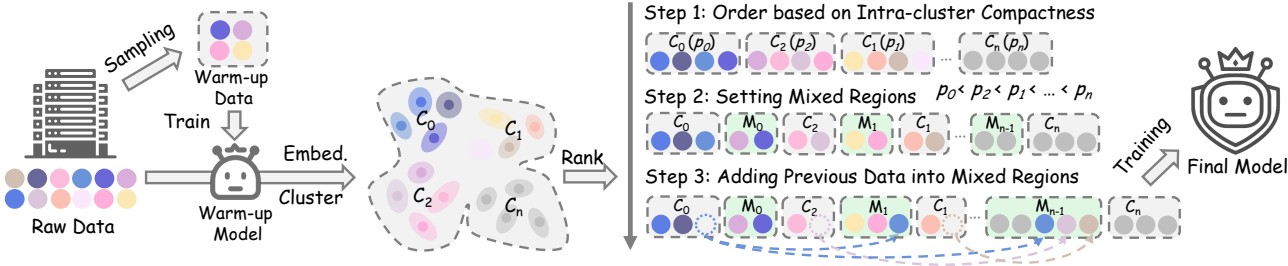

*Figure 2.* Pipeline of the proposed EP-Order paradigm. Training proceeds cluster by cluster, sorted by intra-cluster compactness. Between successive clusters, we insert mixed regions that interleave examples from both sides to reduce gradient conflicts at transitions and to mitigate catastrophic forgetting.

Decompose each gradient into magnitude and direction:

$$a_i \triangleq \|g_i\| \geq 0, u_i \triangleq \begin{cases} g_i/\|g_i\| \in \mathbb{S}^{D-1}, & g_i \neq 0, \\ \text{any } u_i \in \mathbb{S}^{D-1}, & g_i = 0, \end{cases} \quad (2)$$

where $\mathbb{S}^{d-1}$ is the unit sphere. Define the total magnitude $S$ and normalized weights $w_i$:

$$S \triangleq \sum_{k=1}^{B} a_k, w_i \triangleq \frac{a_i}{S} (\geq 0), \quad \sum_{i=1}^{B} w_i = 1. \quad (3)$$

The batch mean direction is

$$m_w \triangleq \sum_{i=1}^{B} w_i u_i \in \mathbb{R}^D. \quad (4)$$

Since $g_i = a_i u_i$,

$$\sum_{i=1}^{B} g_i = \sum_{i=1}^{B} a_i u_i = \left( \sum_{i=1}^{B} a_i \right) \sum_{i=1}^{B} \frac{a_i}{\sum_{k=1}^{B} a_k} u_i = S\, m_w. \quad (5)$$

We define the *Efficiency Ratio* (ER) as

$$\text{ER} \triangleq \frac{\left\| \sum_{i=1}^{B} g_i \right\|}{\sum_{i=1}^{B} \|g_i\|} = \frac{\|S m_w\|}{S} = \|m_w\| \in [0, 1]. \quad (6)$$

Thus, ER depends only on the angular configuration $\{u_i\}$ and weights $\{w_i\}$. In particular, ER $= 1$ iff all $u_i$ are colinear and aligned, and ER $= 0$ iff $m_w = 0$. For a single update step, the directional progress along the consensus direction $\hat{m}_w \triangleq m_w/\|m_w\|$ (when $m_w \neq 0$) is

$$\sum_{i=1}^{B} \langle g_i, \hat{m}_w \rangle = S \langle m_w, \hat{m}_w \rangle = S\|m_w\| = S\, \text{ER}, \quad (7)$$

Hence, larger ER yields larger effective progress per step, accelerating IT. Empirically, grouping semantically similar samples increases ER relative to random interleaving (0.71 vs. 0.54), motivating our group-wise ordering scheme.

## 3.3. Overview

To obtain an effective training data order for IT, EP-Order comprises four stages: (1) warm-up training to obtain embeddings, followed by embedding-based clustering; (2) cluster sorting by intra-cluster compactness; (3) insertion of mixed regions between adjacent clusters; and (4) augmentation of mixed regions with samples from earlier clusters. The resulting order is used to fine-tune multimodal LLMs (*e.g.*, LLaVA (Liu et al., 2024a)) or LLMs (*e.g.* Qwen2.5 (Qwen Team, 2025)). The pipeline of our EP-Order was illustrated in Fig. 2. We now delve into the details of each part.

## 3.4. EP-Order Details

**Warm-up model training.** To efficiently compute sample embeddings for $x_i = (T_i, I_i)$, where $T_i$ denotes the text prompt and $I_i$ denotes the paired image when available, we train a warm-up model $M_{\text{warm}}$ on a small (*e.g.*, 10%), randomly sampled subset of the instruction data. Such a warm-up process equips the $M_{\text{warm}}$ with basic instruction-following while avoiding overfitting to the full data distribution. Note that both the warm-up model and the model for IT share the same architecture and initialization.

**Embedding-based clustering.** We obtain embeddings by forwarding $x_i$ through $M_{\text{warm}}$ and extracting the output of the $\theta$-th decoder layer, followed by token-wise average pooling (TAP):

$$e_i = TAP(M_{\text{warm}}^{(\theta)}(T_i, I_i)) \in \mathbb{R}^{d_e}. \quad (8)$$

After that, we collected an embedding set $E = \{e_1, \ldots, e_t\}$, each embedding $e_i$ corresponding to the $x_i$. We apply UMAP (McInnes et al., 2018) for dimensionality reduction and then HDBSCAN (McInnes & Healy, 2017) to produce $n + 1$ clusters. Compared with another popular cluster technique, $k$-means (McQueen, 1967), HDBSCAN handles variable-density clusters without requiring the number of clusters to be specified. More details of clustering are supplied in the Appendix A.

**Ordering by intra-cluster compactness.** Following cur-

*Table 1.* Full fine-tuning performance of seven data-ordering paradigms and our EP-Order on LLaVA-Vicuna-7B and LLaVA-Vicuna-13B. Average scores are computed over 12 normalized sub-benchmarks. The best result in each column is highlighted in **bold**.

| Method | GQA | MM-Vet | MMBench | | MME | | POPE | | | ScienceQA | | TextVQA | Average |
|---|---|---|---|---|---|---|---|---|---|---|---|---|---|
| | | | en | cn | MME-P | MME-C | adv | pop | rand | SQA | SQA-I | | |
| *LLaVA-Vicuna-7B* | | | | | | | | | | | | | |
| Random | 62.01 | 26.83 | 65.65 | 57.47 | 1464.8 | 301.1 | 84.92 | 86.51 | 87.96 | 70.29 | 68.42 | 57.67 | 64.88 |
| Token Length | 61.29 | 26.74 | 62.76 | 55.07 | 1337.7 | 269.3 | 81.77 | 82.90 | 84.10 | 69.56 | 68.32 | 53.93 | 62.25 |
| Gradient | 59.58 | 26.88 | 63.61 | 54.73 | 1413.2 | 245.4 | 84.83 | 86.56 | 87.39 | 69.28 | 67.53 | 57.31 | 63.25 |
| Loss | 57.98 | 25.32 | 21.94 | 27.75 | 1386.6 | 251.8 | 76.87 | 77.09 | 77.56 | 62.60 | 54.93 | 48.16 | 52.58 |
| Attention Variance (Kim & Lee, 2024) | 59.21 | 22.20 | 61.73 | 54.04 | 1345.1 | 288.2 | 82.21 | 83.72 | 84.70 | 69.56 | 67.72 | 53.03 | 61.78 |
| PDPC (Zhang et al., 2025b) | 61.31 | **29.82** | 64.97 | 57.65 | 1475.6 | 302.1 | 84.85 | 86.06 | 87.73 | 71.00 | 69.01 | **58.09** | 65.17 |
| Bloom's Taxonomy (Ranaldi et al., 2024) | 60.79 | 26.28 | 61.99 | 54.30 | 1475.8 | 297.5 | 83.69 | 85.72 | 87.50 | 68.50 | 65.79 | 52.18 | 63.14 |
| EP-Order | **62.15** | 29.13 | **67.69** | **61.25** | **1512.3** | **331.1** | **85.32** | **86.98** | **88.02** | **71.26** | **69.31** | 57.46 | **66.30** |
| *LLaVA-Vicuna-13B* | | | | | | | | | | | | | |
| Random | **63.11** | 32.06 | 67.26 | 61.08 | 1496.1 | 288.9 | 84.64 | 86.48 | 87.40 | 73.78 | 71.64 | 59.86 | 66.52 |
| Token Length | 62.58 | 29.45 | 66.58 | 57.22 | 1460.7 | 288.6 | 82.47 | 83.90 | 84.73 | **74.68** | **72.34** | 56.57 | 64.97 |
| Gradient | 60.40 | 28.58 | 63.52 | 58.25 | 1527.7 | 296.8 | 84.99 | **86.95** | 86.42 | 72.53 | 69.96 | 59.80 | 65.41 |
| Loss | 59.33 | 28.81 | 50.09 | 50.43 | 1485.8 | 337.1 | 78.01 | 78.85 | 79.11 | 57.63 | 62.75 | 52.75 | 59.53 |
| Attention Variance (Kim & Lee, 2024) | 60.34 | 27.16 | 63.35 | 54.12 | 1385.0 | 290.7 | 82.91 | 84.63 | 85.52 | 73.85 | 71.94 | 51.40 | 63.40 |
| PDPC (Zhang et al., 2025b) | 62.56 | 32.06 | 66.16 | 59.54 | 1546.7 | 326.8 | 84.16 | 85.58 | 86.79 | 73.66 | 71.44 | 60.47 | 66.72 |
| Bloom's Taxonomy (Ranaldi et al., 2024) | 61.71 | 28.81 | 65.56 | 59.88 | 1500.8 | 301.6 | **85.02** | 85.56 | **87.67** | 72.68 | 70.84 | 58.02 | 65.71 |
| EP-Order | 62.65 | **33.26** | **68.37** | **62.20** | 1547.3 | **338.9** | 84.93 | 86.60 | 87.49 | 74.13 | 71.15 | **60.96** | **67.62** |

*Table 2.* LoRA-based fine-tuning performance of seven data-ordering paradigms and our EP-Order on LLaVA-Vicuna-7B. Average scores are computed over 12 normalized sub-benchmarks. The best result in each column is highlighted in **bold**.

| Method | GQA | MM-Vet | MMBench | | MME | | POPE | | | ScienceQA | | TextVQA | Average |
|---|---|---|---|---|---|---|---|---|---|---|---|---|---|
| | | | en | cn | MME-P | MME-C | adv | pop | rand | SQA | SQA-I | | |
| Random | 61.62 | 26.33 | 65.14 | 56.44 | 1472.7 | 270.7 | 84.61 | 86.09 | 87.08 | 69.75 | **69.91** | 58.09 | 64.38 |
| Token Length | 60.10 | 25.96 | 62.76 | 53.61 | 1453.1 | 263.2 | **85.47** | **87.71** | **88.05** | 69.82 | 68.52 | **58.12** | 63.81 |
| Gradient | 60.38 | 26.15 | 63.18 | 53.95 | 1441.4 | 247.9 | 85.35 | 87.20 | 87.97 | 69.91 | 68.27 | 57.76 | 63.60 |
| Loss | 59.14 | 23.53 | 56.46 | 47.94 | 1396.9 | 267.9 | 76.58 | 77.13 | 77.35 | 67.32 | 64.20 | 46.05 | 58.25 |
| Attention Variance (Kim & Lee, 2024) | 60.88 | **31.15** | 56.97 | 49.83 | **1508.7** | 273.6 | 83.34 | 86.17 | 87.68 | 67.96 | 65.10 | 55.88 | 62.88 |
| PDPC (Zhang et al., 2025b) | 61.35 | 29.95 | 63.69 | 54.04 | 1479.5 | 278.2 | 84.36 | 85.75 | 86.79 | 69.70 | 69.21 | 57.73 | 64.28 |
| Bloom's Taxonomy (Ranaldi et al., 2024) | 61.02 | 26.83 | 59.27 | 53.01 | 1450.5 | 282.5 | 84.25 | 86.56 | 87.82 | 69.46 | 67.53 | 56.42 | 63.33 |
| EP-Order | **61.96** | 30.73 | **66.24** | **57.65** | 1493.4 | **337.5** | 84.85 | 86.44 | 87.17 | **71.21** | 68.82 | 57.24 | **65.76** |

riculum learning, we need to calculate the difficulty level for each cluster for ranking. To achieve that, for each cluster $C_j$ with $k_j$ samples, we randomly sample $min(10,000, k_j)$ samples from each cluster to represent the cluster and compute the mean pairwise Euclidean distance among its embeddings as a compactness score $p_j$. The lower mean pairwise Euclidean distance suggests that embeddings within the cluster are closer to each other, and therefore induce lower conflict during model learning. We sort clusters in ascending order of $p_j$, yielding the original training order $O = \{C_0, C_1, \ldots, C_n\}$ with $p_0 < p_1 < \cdots < p_n$. Within each $C_i$, the training order is randomly shuffled.

**Mixed regions between adjacent clusters.** While within-cluster training exhibits low conflict, transitions between clusters always induce sharp gradient changes, resulting in high conflict. Such conflict introduces the risk of unstable optimization, degraded generalization on previously seen data, and even catastrophic forgetting when the model rapidly shifts its focus to the new cluster without sufficient overlap in gradient directions. To smooth transitions, we build a mixed region $M_i$ between $C_i$ and $C_{i+1}$ by selecting equal-sized mixed sets $c_i \subset C_i$ and $c_{i+1} \subset C_{i+1}$ and

shuffling their union. Therefore, the training order becomes $O = \{C_0, M_0, C_1, M_1, ..., M_{n-1}, C_n\}$. After insertion, the effective size of each $C_i$ is reduced accordingly. Note that using HDBSCAN for clustering can bring some samples regarded as noise samples. We randomly shuffle these samples and uniformly assign them to the mixed regions, avoiding information loss.

**Adding previous data into mixed regions.** Clustering can concentrate related knowledge, increasing the risk of forgetting when moving forward. To counter this, we further augment each $M_i$ with a small, randomly sampled buffer from previously learned clusters. For example, for $M_{n-1} = c_{n-1} \cup c_n$, we add buffer samples $\{b_0, \ldots, b_{n-2}\}$ drawn from $\{C_0, \ldots, C_{n-2}\}$, producing $M'_{n-1} = \{c_{n-1}, c_n, b_0, \ldots, b_{n-2}\}$. Applying this augmentation to all mixed regions yields the final order $O_{final} = \{C_0, M'_0, C_1, M'_1, ..., M'_{n-1}, C_n\}$. For $M_0$, no previous-cluster buffer is added, so $M'_0 = M_0$.

**Model training.** As a static data ordering paradigm, the EP-Order is performed prior to the IT to obtain $O_{final}$. We train the target model using its tailored training data order without further changes to the standard IT pipeline.

*Table 3.* Ablation study of Group-based Order, Mixed Region, and Replay Buffer Data components on 12 sub-benchmarks. The Random paradigm serves as the baseline. Backbone: LLaVA-Vicuna-7B.

| Method | GQA | MM-Vet | MMBench | | MME | | POPE | | | ScienceQA | | TextVQA | Average |
| | | | en | cn | MME-P | MME-C | adv | pop | rand | SQA | SQA-I | | |
| --- | --- | --- | --- | --- | --- | --- | --- | --- | --- | --- | --- | --- | --- |
| Random (Baseline) | 62.01 | 26.83 | 65.65 | 57.47 | 1464.8 | 301.1 | 84.92 | 86.51 | 87.96 | 70.29 | 68.42 | 57.67 | 64.88 |
| + Group-based Order | 62.12 | 27.66 | 65.31 | 57.56 | 1443.3 | 289.3 | 84.06 | 85.61 | 87.32 | 70.83 | 69.96 | 57.63 | 64.70 |
| + Mixed Region | 62.03 | 29.31 | 67.09 | 59.79 | 1477.9 | 300.4 | 84.41 | 86.04 | 87.21 | 71.49 | 70.20 | 57.32 | 65.53 |
| + Replay Buffer Data | 62.15 | 29.13 | 67.69 | 61.25 | 1512.3 | 331.1 | 85.32 | 86.98 | 88.02 | 71.26 | 69.31 | 57.46 | 66.30 |
| EP-Order w/o Group-based Order | 62.21 | 28.58 | 66.07 | 59.88 | 1481.9 | 267.5 | 84.41 | 85.69 | 87.44 | 69.49 | 66.63 | 57.11 | 64.59 |

*Table 4.* Performance of four $k$-means–based orderings and one HDBSCAN-based ordering on seven benchmarks. **Academic**: academic-task-oriented benchmarks (GQA, ScienceQA, and TextVQA). **IF**: instruction-following multimodal LLM benchmarks (MM-Vet, MMBench, MME, and POPE). **Average**: mean score over all 12 sub-benchmarks. The best result in each column is highlighted in **bold**. Backbone: LLaVA-Vicuna-7B.

| Method | Academic | IF | Average |
| --- | --- | --- | --- |
| $k$-means ($k$=5) | 63.67 | 64.37 | 64.14 |
| $k$-means ($k$=10) | 65.00 | 65.96 | 65.64 |
| $k$-means ($k$=15) | 64.89 | 65.85 | 65.53 |
| $k$-means ($k$=20) | 64.51 | 65.25 | 65.00 |
| HDBSCAN | **65.05** | **66.92** | **66.30** |

# 4. Experiment

## 4.1. Experimental Setup

**Multimodal data.** We adopted the fine-tuning data from the LLaVA-1.5 (Liu et al., 2024a), a mixed multimodal instruction-following corpus (VQA, OCR, region-level VQA, visual conversation, and language-only dialogue), totaling ∼665K samples. The evaluation data spans seven benchmarks, including academic-task-oriented datasets, GQA (Hudson & Manning, 2019), ScienceQA (Lu et al., 2022), TextVQA (Singh et al., 2019), and instruction-following multimodal LLM suites, including MM-Vet (Yu et al., 2024), MMBench (Liu et al., 2024b), MME (Fu et al., 2023), and POPE (Li et al., 2023). More details are provided in the Appendix B.

**Multimodal Models.** We adopt LLaVA-1.5-7B and LLaVA-1.5-13B as multimodal LLM backbones, whose language decoders are Vicuna-7B and Vicuna-13B (Zheng et al., 2023). The visual encoder is a pretrained ViT-Large (Radford et al., 2021).

**Training.** We followed LLaVA-1.5 (Liu et al., 2024a) for instruction fine-tuning. For full fine-tuning (vision encoder frozen), we used AdamW, batch size 128, and trained for 1 epoch with a cosine schedule. The learning rate warms from 0.0 to $2 \times 10^{-5}$ over 3% of steps, then decays cosinely. We used greedy decoding for evaluation to ensure reproducibility. For *LoRA-based* fine-tuning (Hu et al., 2022), we set rank $= 128$ and $alpha = 256$. Hyperparameters

match full fine-tuning except that the base learning rate is $2 \times 10^{-4}$ while the MLP projection layer uses $2 \times 10^{-5}$. All experiments run on $8 \times$ GPUs.

**Baselines.** We compared EP-Order with Random and several data-ordering paradigms: (1) **Random**: The training order is determined by a random shuffle. (2) **Token Length**: Samples are sorted by the total token count of the text prompt and image (from small to large). (3) **Gradient**: Each sample's aggregated $L_2$ norm over all trainable parameters is computed using the warm-up model (trained on 10% of the data), and samples are sorted by gradient norm (from small to large). (4) **Loss**: Similar to Gradient, but using the per-sample loss from the warm-up model (from small to large). (5) **Attention Variance** (Kim & Lee, 2024), **PDPC** (Zhang et al., 2025b), and **Bloom's Taxonomy** (Ranaldi et al., 2024) are implemented following their official settings. Details are shown in Appendix C.

## 4.2. Comparison with Data Ordering Paradigms

We comprehensively compare EP-Order with recent data-ordering paradigms, including Random (the most widely used paradigm), Token Length, Gradient, Loss, Attention Variance (Kim & Lee, 2024), PDPC (Zhang et al., 2025b), and Bloom's Taxonomy (Ranaldi et al., 2024), on seven multimodal benchmarks. LLaVA-Vicuna-7B and LLaVA-Vicuna-13B (Liu et al., 2024a) serve as backbones. Both full fine-tuning (vision encoder frozen) and LoRA-based fine-tuning are evaluated.

**Full fine-tuning.** Results are reported in Table 1. We observed that, despite its implementation simplicity and zero overhead for data arrangement, the Random paradigm achieves the best or near-best generalization across 12 sub-benchmarks, outperforming most prior ordering methods, with PDPC being the closest competitor. This highlights the limited robustness of existing techniques based on curriculum learning when transferred to challenging multimodal LLM benchmarks and underscores the difficulty of directly porting conventional curriculum techniques to multimodal LLMs. In contrast, EP-Order consistently improves performance across most benchmarks. With LLaVA-Vicuna-7B, EP-Order achieves the best results on 10 out of 12 sub-benchmarks and surpasses Random and the second-best

*Table 5.* Performance of five sorting strategies based on the proposed compactness scores. In the folding strategy, clusters are divided into $n$ groups, and each group is ordered from 'easy' to 'hard'. Backbone: LLaVA-Vicuna-7B.

| Method | GQA | MM-Vet | MMBench | | MME | | POPE | | | ScienceQA | | TextVQA | Average |
|---|---|---|---|---|---|---|---|---|---|---|---|---|---|
| | | | en | cn | MME-P | MME-C | adv | pop | rand | SQA | SQA-I | | |
| Random Shuffle | 61.12 | 28.26 | 64.71 | 57.90 | 1481.7 | **348.2** | 84.99 | 86.69 | 87.83 | **71.42** | 69.36 | 57.16 | 65.59 |
| Folding ($n$=5) (Dai et al., 2025) | 61.77 | 29.59 | 64.71 | 58.76 | 1457.9 | 262.9 | 84.64 | 86.37 | 87.33 | 71.21 | **70.15** | 58.52 | 64.90 |
| Folding ($n$=3) (Dai et al., 2025) | **62.21** | 29.13 | 65.82 | 56.36 | 1474.6 | 282.5 | 85.10 | **87.07** | 87.99 | 70.60 | 69.51 | 58.18 | 65.08 |
| Hard → Easy | 62.17 | **30.92** | 65.99 | 56.44 | **1514.8** | 308.2 | 85.31 | 86.66 | 87.90 | 70.01 | 68.86 | **58.64** | 65.60 |
| Easy → Hard | 62.15 | 29.13 | **67.69** | **61.25** | 1512.3 | 331.1 | **85.32** | 86.98 | **88.02** | 71.26 | 69.31 | 57.46 | **66.30** |

method (PDPC) by 1.42 and 1.13 average performance, respectively. When scaling from LLaVA-Vicuna-7B to LLaVA-Vicuna-13B, we observe a similar pattern: Random remains a strong (and sometimes competitive) baseline, yet EP-Order attains the best performance on 6 out of 12 sub-benchmarks and achieves the highest average score.

**LoRA-based fine-tuning.** We further compared EP-Order with competing paradigms under the LoRA-based fine-tuning setting using LLaVA-Vicuna-7B as the backbone. The results, presented in Table 2, show that EP-Order achieves the best average performance, outperforming the second-best method (Random) by 1.38 average points.

Overall, prior ordering paradigms often underperform the widely used Random baseline, and naively applying curriculum-learning signals (*e.g.*, loss-based ordering) can even degrade performance. By contrast, EP-Order is well-suited to multimodal LLM benchmarks and demonstrates consistent gains under both full and LoRA-based fine-tuning.

### 4.3. Ablation Studies

We conducted ablations on the key components of EP-Order: Group-based ordering, mixed regions (MR), and replay buffer data (RBD). These three strategies are incrementally added to obtain the final EP-Order. Results are shown in Table 3 using LLaVA-Vicuna-7B, with Random as the baseline. Replacing Random with group-based ordering alone leads to mixed effects: some benchmarks (*e.g.*, MM-Vet, ScienceQA) improve, while others (*e.g.*, MME, POPE) degrade, resulting in comparable or slightly worse average performance overall. This behavior aligns with our intuition. Although grouping semantically similar samples accelerates learning within each group, it also introduces sharp gradient shifts at group boundaries and exacerbates the forgetting of group-specific knowledge, which can ultimately harm overall performance. After inserting MR with a small size, we are able to observe consistent improvements on most benchmarks, with only minor drops on GQA and TextVQA. This indicates that smoothing transitions between clusters reduces optimization risk induced by sharp changes in gradient directions. Finally, introducing RBD yields the full EP-Order and achieves the best average results, underscoring

the importance of buffer samples for alleviating catastrophic forgetting under group-based training. To directly examine whether the gains mainly come from MR and RBD rather than meaningful grouping, we further introduce a control experiment. Specifically, we keep the same 15-group schedule, the same group-size distribution, and the same MR/RBD design, but replace embedding-based grouping with random partitions. This variant, denoted as "EP-Order w/o Group-based Order," achieves an average performance of 64.59, which is comparable to Random (64.88) but clearly lower than full EP-Order (66.30). This result confirms that MR and RBD alone cannot explain the improvement; meaningful group-based ordering is essential for unlocking their effectiveness. In summary, group-based ordering provides a strong baseline for multi-task learning but has inherent weaknesses. Introducing a small size of mixed regions helps address these issues and substantially unlocks its potential.

### 4.4. Different Clustering Strategies

The default clustering method in EP-Order is HDBSCAN with automatically determined cluster count. We also evaluated $k$-means clustering with four choices of cluster numbers ($k \in \{5, 10, 15, 20\}$), using LLaVA-Vicuna-7B as the backbone. Results are reported in Table 4. We find that too few clusters (*e.g.*, $k = 5$) or too many clusters (*e.g.*, $k = 20$) hurt performance, in some cases significantly underperforming Random. Intuitively, too few clusters fail to produce highly similar, low-conflict groups, while too many clusters exacerbate catastrophic forgetting. HDBSCAN automatically yields 15 clusters in our setting and achieves the best performance across metrics. Moreover, even when controlling for the same number of clusters ($k = 15$), HDBSCAN consistently outperforms $k$-means. We attribute this difference to the distinct inductive biases of the clustering algorithms. HDBSCAN favors forming compact, density-connected clusters, which better align with the goal of constructing low-conflict training groups. In contrast, $k$-means enforces a fixed partitioning of the representation space and may split a semantically coherent cluster or forcibly assign transition-region samples to a centroid-driven cluster (see Appendix A for visualization). Such artifacts reduce intra-cluster consistency and weaken the effectiveness of cluster-based ordering.

*Table 6.* Full fine-tuning performance of six data-ordering paradigms and EP-Order on Qwen3-4B. Average scores are computed over four benchmarks, and the best result in each column is highlighted in **bold**.

| Method | ARC | BBH | GSM | MMLU | Average |
|---|---|---|---|---|---|
| Random | 71.86 | 72.29 | 82.71 | **72.39** | 74.81 |
| Token Length | 81.83 | 72.44 | 70.89 | 71.39 | 74.14 |
| Gradient | 81.36 | 73.45 | 61.26 | 31.91 | 62.00 |
| Loss | 79.66 | **74.71** | 83.47 | 32.58 | 67.61 |
| Attention Variance (Kim & Lee, 2024) | 85.80 | 74.33 | 64.14 | 71.45 | 73.93 |
| PDPC (Zhang et al., 2025b) | 70.85 | 74.22 | 75.97 | 71.63 | 73.17 |
| EP-Order | **86.44** | 72.15 | **88.10** | 71.33 | **79.51** |

*Table 7.* Performance of the Random paradigm and EP-Order for hybrid thinking training. Models are evaluated on MATH500, AIME24, and GPQA under think and no-think evaluation modes.

| Method | Data | MATH500 | | AIME24 | | GPQA | |
|---|---|---|---|---|---|---|---|
| | | Think | No-think | Think | No-think | Think | No-think |
| | No-think | - | 63.34 | - | 5.33 | - | **28.79** |
| Random | Think | 69.44 | - | 10.00 | - | 27.58 | - |
| | No-think + Think | **71.54** | 63.44 | 11.67 | 5.33 | 28.33 | 25.71 |
| EP-Order | No-think + Think | 71.52 | **63.60** | **13.67** | **6.33** | **28.84** | 27.12 |

## 4.5. Different Ordering Strategies within EP-Order

We also investigated alternative ordering strategies based on the compactness scores, as summarized in Table 5 (LLaVA-Vicuna-7B backbone). The cluster order is determined either in ascending (Easy → Hard) or descending (Hard → Easy) order according to the proposed compactness scores. We also adopted the folding strategy (Dai et al., 2025), which divides clusters into groups and applies an Easy → Hard ordering within each group. For completeness, we also considered randomly shuffling the cluster order. The results show that both folding variants (with 5 or 3 groups) perform poorly on the benchmarks, yielding only marginal gains over Random. In contrast, straightforward curricula that order clusters globally from Hard → Easy or Easy → Hard, even a random order (Random Shuffle), perform better than folding. Among these, sorting clusters from Easy → Hard by compactness achieves the best overall performance, and we therefore adopted this setting as the default configuration of EP-Order. The inferior performance of the folding strategies further supports our hypothesis that abrupt changes in gradient directions (when switching between folds) can hinder optimization. Although mixed regions partially alleviate this issue, they are insufficient to fully compensate for the instability introduced by fold-wise ordering.

## 4.6. Training Efficiency for EP-Order

We verified that the group-based ordering yields a higher ER than the Random paradigm, indicating that EP-Order enables more efficient training. To further examine whether a higher ER translates into faster loss optimization, we visualize loss differences in Fig. 3, defined as $\Delta\mathcal{L}_t = \mathcal{L}_t - \mathcal{L}_{t-1}$, $t$: Step, $\mathcal{L}$: Loss function, *i.e.*, the loss at the current step

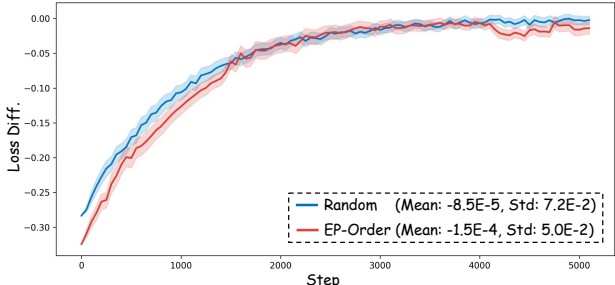

*Figure 3.* Visualization of loss differences under the Random paradigm and EP-Order during training. Loss difference denotes the change in loss between two consecutive training steps (stride=50). An exponential moving average with $\alpha = 0.05$ is applied for smoothing. Mean and Std are computed over loss differences (stride=1). Lower loss differences imply faster learning.

minus that at the previous step (computed between consecutive sampled training steps). As shown in Fig. 3, EP-Order (red) generally exhibits more negative loss differences than Random (blue), particularly at the beginning and near the end of training, indicating faster loss reduction. Overall, EP-Order achieves a more negative mean loss difference (approximately $2\times$ smaller in value than Random) with a smaller standard deviation, suggesting that the group-based ordering enables both faster and more stable optimization.

## 4.7. EP-Order for More Text-only Benchmarks

To further evaluate the generalization ability of EP-Order, we conduct experiments with Qwen3-4B (Yang et al., 2025) on Tulu-3-SFT (Lambert et al., 2025) and evaluate the resulting models on ARC (Clark et al., 2018), Big-Bench-Hard (BBH) (Suzgun et al., 2023), GSM8k (GSM) (Cobbe et al., 2021), and MMLU (Hendrycks et al., 2021), comparing six ordering paradigms. As shown in Table 6, the conclusions are consistent with the multimodal results: Random remains strong; prior methods are often unbalanced (e.g., Attention Variance improves ARC but drops substantially on GSM); and EP-Order achieves the best results on ARC (86.44) and GSM (88.10) while remaining competitive on BBH (72.15) and MMLU (71.33). Overall, EP-Order improves the average score from 74.81 to 79.51, further showing that its effectiveness is not specific to multimodal IT.

## 4.8. EP-Order in Hybrid Thinking Scenarios

EP-Order is flexible and broadly applicable. We further applied it to a text-only hybrid-thinking scenario, which exhibits non-negligible optimization conflict when jointly training think and no-think samples. Following (Wang et al., 2025), we used 80K examples from the OpenR1-Math (Hugging Face, 2025) default subset and construct a no-think version by removing the chain-of-thought for half of the samples (40K). We fine-tuned Qwen2.5-7B-Instruct

(Qwen Team, 2025) for one epoch. More training details and experimental results are provided in Appendix E. The resulting models are evaluated on MATH500 (Lightman et al., 2024), AIME24[2], and GPQA (Rein et al., 2024) under both think and no-think evaluation modes. We compared pure no-think training, pure think training, and hybrid training with Random ordering. As shown in Table 7, naively mixing think and no-think data does not bring significant performance improvements and can even result in a performance decrease on the GPQA, as its no-think accuracy drops from 28.79 to 25.71. With EP-Order, hybrid think training achieves substantial improvements across most metrics. The GPQA no-think accuracy increases from 25.71 to 27.12, and the AIME24 think accuracy increases from 11.67 to 13.67, which suggests that embedding-proximity-based ordering provides a promising perspective for mitigating interference in hybrid think/no-think training.

## 5. Conclusion

In this paper, we propose a new data ordering paradigm for instruction tuning (IT) of multimodal LLMs and LLMs. Unlike prior approaches that assign per-sample scores and sort all examples accordingly, EP-Order partitions the dataset into multiple groups and operates at the group level. Compared with the widely used Random paradigm, our group-based EP-Order leverages richer semantic information, substantially increasing the utilization of useful gradient signals within a mini-batch caused by optimization conflicts. We evaluated EP-Order on seven popular instruction-following multimodal LLM benchmarks and demonstrated its effectiveness and robustness across both full and LoRA-based fine-tuning. Moreover, by transferring EP-Order to a hybrid thinking text-only scenario, we showed that it can effectively alleviate intra- and inter-group conflicts between think and no-think samples, achieving superior performance over random ordering. In future work, we plan to extend EP-Order to dynamic training schedules under multi-epoch settings and to explore its integration with broader families of curriculum learning and data selection strategies.

## Acknowledgements

This research was conducted at ByteDance. We are grateful to our colleagues at ByteDance for their valuable suggestions and discussions.

## Impact Statement

EP-Order may have a positive impact by making instruction tuning more stable and effective through improved data ordering, and by helping researchers better understand the relationship between data order and optimization behavior. At the same time, it introduces additional preprocessing overhead, depends on the quality of learned embeddings and clustering results, and may involve a trade-off between optimization efficiency and final generalization.

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

## A. Details of Clustering

For each sample, we first extract the hidden states from the $\theta$-th decoder layer and apply token-wise average pooling to obtain a single embedding vector, which is more compact and clustering-friendly. We then reduce the embedding dimension from $d_e$ to $d_{low}$ (*e.g.*, from 4096 to 10 for LLaVA-Vicuna-7B) using UMAP, obtaining low-dimensional embeddings $e_{low} \in \mathbb{R}^{d_{low}}$. HDB-SCAN is subsequently applied to $\{e_{low}\}$ to produce multiple clusters. The main hyperparameters are set as: 'min_cluster_size'=5000, 'min_samples'=100, 'metric'='euclidean', 'cluster_selection_method'='eom', 'cluster_selection_epsilon'=0.30.

We further visualize the resulting clusters using t-SNE, as shown in Figure 4, and compare two coloring schemes. **Clustering-based coloring**: each color corresponds to a cluster produced by $k$-means or HDBSCAN. **Sub-dataset-based coloring**: each color corresponds to a sub-dataset of the LLaVA-1.5 training data, following the partition provided by TIVE (Liu et al., 2025). We randomly sample 5,000 embeddings per category for visualization. The plots reveal that, under embedding-proximity clustering, sub-datasets are not good surrogates for semantic groups: many samples from different sub-datasets lie close in embedding space. Moreover, Fig. 4 shows that the $k$-means-based partitioning with a pre-defined number of clusters can split semantically coherent groups, preventing highly related samples from being learned together. This fragmentation likely explains why $k$-means underperforms HDBSCAN, even when using the same number of clusters. We also tried an EP-Order variant that uses sub-datasets as groups. Specifically, we compute a compactness score for each sub-dataset and sort sub-datasets from 'easy' to 'hard' based on these scores; the remaining steps mirror those of EP-Order. As reported in Table 8, sub-dataset–based ordering leads to large performance drops compared with cluster-based EP-Order on most benchmarks, with the exception of POPE-pop and POPE-rand, where it yields performance gains. We attribute the poor overall performance of sub-dataset–based ordering to catastrophic forgetting: some types of knowledge appear predominantly in a single sub-dataset that is learned early in training. Although EP-Order includes buffer samples to mitigate this effect, the buffer size is insufficient to fully counteract it. In contrast, cluster-based ordering groups samples by embedding proximity rather than dataset identity, which spreads related knowledge across clusters and substantially reduces this risk.

## B. Details of Benchmarks

**GQA** (Hudson & Manning, 2019) is a large-scale visual question answering benchmark constructed from scene graphs over real-world images in Visual Genome. It eval-uates fine-grained visual reasoning, including object and attribute recognition, spatial relations, and multi-step compositional reasoning.

**ScienceQA** (Lu et al., 2022) is a multimodal science question answering benchmark with multiple-choice questions spanning natural science, language science, and social science. It assesses models' abilities to understand multimodal content, leverage external knowledge, and perform complex scientific reasoning, often with chain-of-thought explanations.

**TextVQA** (Singh et al., 2019) is a visual question answering dataset that requires reading and reasoning about scene text in images. It evaluates a model's ability to integrate visual perception with OCR-based text recognition and linguistic reasoning to answer natural-language questions.

**MM-Vet** (Yu et al., 2024) is an evaluation benchmark targeting challenging multimodal tasks for large multimodal models, organized around six core vision–language capabilities and sixteen integrated capability combinations. It measures integrated skills such as OCR, spatial and mathematical reasoning, commonsense, and world knowledge in open-ended image–question settings.

**MMBench** (Liu et al., 2024b) is a systematically designed bilingual benchmark for robust and holistic evaluation of vision–language models. It is available in both English (en) and Chinese (cn) versions, enabling cross-lingual assessment of models' comprehensive multimodal understanding.

**MME** (Fu et al., 2023) is a comprehensive evaluation suite for multimodal large language models that measures both perception and cognition across 14 subtasks using manually designed instruction–answer pairs. It assesses general vision–language competence, including recognition of objects, attributes, and text, as well as higher-level reasoning under concise, standardized instructions.

**POPE** (Li et al., 2023) is a benchmark and protocol for assessing object hallucination in large vision–language models. Built on images with ground-truth object annotations (*e.g.*, COCO) and yes/no probes such as 'Is there a {object} in the image?', it quantifies models' tendency to hallucinate non-existent objects and provides stable, flexible metrics for hallucination rates across datasets and models.

## C. Details of Baselines

**Attention Variance** (Kim & Lee, 2024) performs data ordering in three steps. *Step 1:* Train a warm-up model. We follow EP-Order and randomly sample 10% of the training data to train this warm-up model. *Step 2:* Compute attention variance for each sample using the official imple-

*Table 8.* Performance of EP-Order when ordering by sub-datasets versus by clusters. Backbone: LLaVA-Vicuna-7B. The best result in each column is highlighted in **bold**.

| Method | GQA | MM-Vet | MMBench | | MME | | POPE | | | ScienceQA | | TextVQA | Average |
|---|---|---|---|---|---|---|---|---|---|---|---|---|---|
| | | | en | cn | MME-P | MME-C | adv | pop | rand | SQA | SQA-I | | |
| EP-Order w/ Sub-datasets | 47.61 | 26.15 | 57.23 | 48.37 | 1451.3 | 259.3 | 84.17 | **87.24** | **89.45** | 62.63 | 60.93 | 43.22 | 59.33 |
| EP-Order w/ Clusters | **62.15** | **29.13** | **67.69** | **61.25** | **1512.3** | **331.1** | **85.32** | 86.98 | 88.02 | **71.26** | **69.31** | **57.46** | **66.30** |

*Table 9.* Time cost of each component for Attention Variance (AV), PDPC, Bloom's Taxonomy, and EP-Order. Time is reported in hours (h).

| Attention Variance | Step 1: Warm-up Training | Step 2: AV Calculation & Sort | | Step 3: Full Training | Total Time |
|---|---|---|---|---|---|
| | ~1.6h | ~4.5h | | ~16h | ~22.1h |
| PDPC | Step 1: Weak Model Training | Step 2: Strong Model Training | Step 3: PD calculation & Sort | Step 4: Full Training | Total Time |
| | ~1.6h | ~16h | ~8h | ~16h | ~41.6h |
| Bloom's Taxonomy | Step 1: LLM Scores | | Step 2: Sort | Step 3: Full Training | Total Time |
| | ~7 days | | ~0.0h | ~16h | ~184h |
| EP-Order | Step 1: Warm-up Training | Step 2: Embedding Extraction | Step 3: Clustering & Sort | Step 4: Full Training | Total Time |
| | ~1.6h | ~3.7h | ~0.3h | ~16h | ~21.6h |

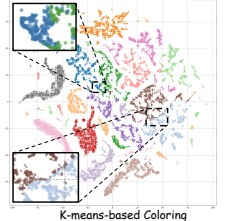 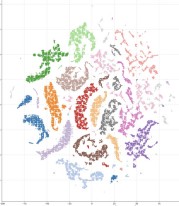 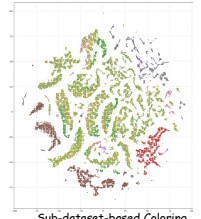

K-means-based Coloring    HDBSCAN-based Coloring    Sub-dataset-based Coloring

*Figure 4.* T-SNE visualizations of clustered embeddings colored based on clustering, obtained by $k$-means, HDBSCAN, or sub-datasets, which consist of the LLaVA-1.5 training data.

mentation[3]. *Step 3:* Sort samples by attention variance from large ('easy') to small ('hard') values and use this order for IT.

**PDPC** (Zhang et al., 2025b) also adopts a three-step pipeline. *Step 1:* Train a weak and a strong model. In our setup, the weak model is the warm-up model trained on 10% of the data, and the strong model is the fully trained model using 100% of the data. We do not follow the original PDPC setting where the weak model is a smaller model (*e.g.*, 1.5B) and the strong model is an equal-size model (*e.g.*, 7B for LLaVA-Vicuna-7B), because no smaller pretrained Vicuna-based decoder is available. Thus, adopting the Early-End strategy is effectively the only feasible choice. Notably, the original paper reports that the two strategies yield very similar performance (54.88 vs. 54.69), suggesting limited sensitivity to this design.

**Bloom's Taxonomy** (Ranaldi et al., 2024) first assigns a Bloom's taxonomy level to each sample and then orders the data accordingly. We design a prompt (shown in Figure 5)

_______________
[3]https://github.com/KoJLabs/StrategicDataOrdering

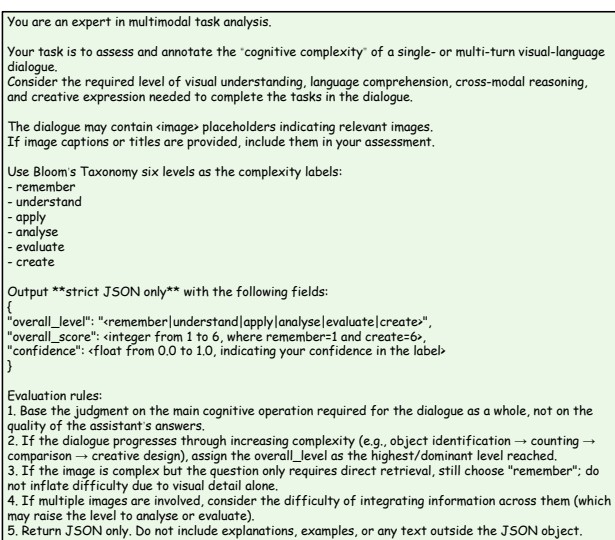

*Figure 5.* Prompt used in Bloom's Taxonomy to select the taxonomy level for multimodal samples.

tailored to multimodal samples and use an SOTA LLM to obtain the taxonomy scores. Samples are then sorted from 'remember' to 'create'. Within each complexity level, we further sort samples by token length from short to long.

# D. Overhead of Data Ordering

Any data-ordering scheme introduces extra overhead to compute ordering signals. We therefore measured the total extra time of EP-Order and three strong competitors: Attention Variance (Kim & Lee, 2024), PDPC (Zhang et al., 2025b), and Bloom's Taxonomy (Ranaldi et al., 2024). Table 9 reports the detailed time cost of each component for all

*Table 10.* Performance of the Random paradigm and EP-Order for hybrid thinking training with 140K training samples. Models are evaluated on MATH500, AIME24, and GPQA under think and no-think evaluation modes.

| Method | Data | MATH500 | | AIME24 | | GPQA | |
|---|---|---|---|---|---|---|---|
| | | Think | No-think | Think | No-think | Think | No-think |
| Random | No-think | - | 61.24 | - | 3.67 | - | 27.78 |
| | Think | 73.78 | - | **14.33** | - | 29.50 | - |
| | No-think + Think | 75.22 | **64.80** | 12.67 | **4.00** | 28.54 | 27.83 |
| EP-Order | No-think + Think | **75.28** | 64.60 | **14.33** | 3.67 | **30.40** | **28.90** |

*Table 11.* Full fine-tuning performance of seven data-ordering paradigms and our EP-Order on LLaVA-LLaMA2-7B. Average scores are computed over 12 normalized sub-benchmarks. The best result in each column is highlighted in **bold**.

| Method | GQA | MM-Vet | MMBench | | MME | | POPE | | | ScienceQA | | TextVQA | Average |
|---|---|---|---|---|---|---|---|---|---|---|---|---|---|
| | | | en | cn | MME-P | MME-C | adv | pop | rand | SQA | SQA-I | | |
| Random | **62.51** | 29.36 | 65.82 | 55.76 | 1496.7 | 281.4 | 84.91 | 86.29 | 87.91 | 69.30 | 67.92 | 56.64 | 64.70 |
| Token Length | 61.04 | 26.06 | 57.23 | 46.65 | 1392.8 | 288.9 | 81.20 | 82.09 | 83.23 | 58.48 | 67.58 | 53.18 | 60.21 |
| Gradient | 61.23 | 29.31 | 65.82 | 57.65 | 1421.9 | 282.5 | 84.52 | 85.96 | 87.07 | **71.49** | 68.47 | 56.21 | 64.51 |
| Loss | 58.90 | **31.01** | 62.50 | 50.52 | 1374.3 | 247.1 | 82.50 | 84.04 | 84.83 | 63.57 | 66.14 | 48.81 | 61.04 |
| Attention Variance (Kim & Lee, 2024) | 59.60 | 24.95 | 60.71 | 54.21 | 1303.6 | 293.2 | 82.69 | 83.39 | 85.07 | 70.24 | 68.32 | 8.17 | 58.27 |
| PDPC (Zhang et al., 2025b) | 46.06 | 29.59 | 59.52 | 51.37 | 1315.7 | 291.1 | 80.43 | 84.73 | 86.87 | 69.32 | 67.53 | 55.80 | 61.12 |
| Bloom's Taxonomy (Ranaldi et al., 2024) | 60.73 | 26.88 | 63.27 | 53.69 | 1471.4 | 313.6 | 83.88 | 86.43 | **88.40** | 69.91 | 67.58 | 54.61 | 64.01 |
| EP-Order | 62.41 | 29.63 | **67.35** | **58.25** | **1519.1** | **320.7** | **85.45** | **87.21** | 88.03 | 70.62 | **69.16** | **57.60** | **65.98** |

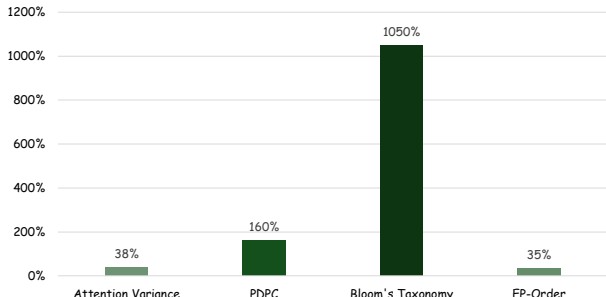

*Figure 6.* Ratio between data ordering overhead and IT training time for Attention Variance, PDPC, Bloom's Taxonomy, and EP-Order under full fine-tuning on LLaVA-Vicuna-7B. All experiments are conducted on 8 × GPUs.

data-ordering paradigms. The results show that the computation of the sorting metric is the dominant contributor to overhead. Our EP-Order is the most efficient: it reuses the output of a single decoder layer as the ordering signal, avoiding repeated passes or additional heavy processing. In contrast, PDPC must train two models to estimate the Perplexity Difference (PD), and Attention Variance requires storing and processing all attention maps. The overhead of Bloom's Taxonomy is largely determined by the degree of concurrency in LLM API calls. The ratios between data ordering overhead and IT training time of these paradigms are shown in Figure 6. Totally, EP-Order incurs the lowest total overhead among these paradigms, adding approximately 35% extra time (5.6h) over Random. EP-Order reduces the additional overhead by 0.5h, 20.0h, and 162.4h compared

with Attention Variance, PDPC, and Bloom's Taxonomy, respectively.

All data ordering paradigms necessarily introduce extra costs for data evaluation, and these overheads are often comparable to, or even exceed, the IT training time. Nevertheless, data re-ordering remains a potential strategy for alleviating data-induced optimization conflicts without modifying training code, making it highly engineering-friendly.

# E. EP-Order on Hybrid Thinking Training

In Section 4.8, we constructed a hybrid thinking dataset with 40K think samples and 40K no-think samples to evaluate the effectiveness of EP-Order. We ensured that the 40K no-think samples were not derived from the corresponding 40K think samples. All experiments were conducted using the LLaMA-Factory framework[4]. In this section, we scale the training set from 80K to 140K samples, consisting of 70K think and 70K no-think examples. Unlike before, we now allow overlap between think and no-think data, *i.e.*, a no-think instance may be constructed from the same original problem as a think instance. The results in Table 10 show that EP-Order provides a more suitable ordering for hybrid thinking training, yielding performance gains on most datasets. The only exceptions are MATH500 (no-think) and AIME24 (no-think), where performance is slightly lower than the Random baseline. Moreover, when comparing with Table 7, we observe that increasing the number of no-think samples from 40K to 70K leads to performance drops when training

---

[4] https://github.com/hiyouga/LLaMA-Factory

*Table 12.* Performance of EP-Order with different mixed set sizes $|c_i|$ and buffer sample sizes $|b_i|$ on three metrics. Here, $|c_i|$ and $|b_i|$ denote the number of mixed and buffer samples per region, respectively.

| Method | Academic | IF | Average |
|---|---|---|---|
| $c_i$=512, $b_i$=256 | 64.97 | 66.15 | 65.75 |
| $c_i$=1024, $b_i$=256 | **65.05** | **66.92** | **66.30** |
| $c_i$=2048, $b_i$=256 | 64.87 | 65.83 | 65.51 |
| $c_i$=1024, $b_i$=128 | 64.85 | 65.23 | 65.10 |
| $c_i$=1024, $b_i$=256 | **65.05** | **66.92** | **66.30** |
| $c_i$=1024, $b_i$=512 | 64.84 | 65.99 | 65.61 |

*Table 13.* Performance of EP-Order using cluster embeddings from the $\theta$-th decoder layer. The best result in each column is highlighted in **bold**. Backbone: LLaVA-Vicuna-7B.

| $\theta$-th decoder layer | Academic | IF | Average |
|---|---|---|---|
| $\theta$=1 | 63.79 | 64.73 | 64.42 |
| $\theta$=7 | 64.52 | 65.31 | 65.04 |
| $\theta$=10 | 64.32 | 65.73 | 65.26 |
| $\theta$=16 | 64.46 | 66.07 | 65.53 |
| $\theta$=32 | **65.05** | **66.92** | **66.30** |

on no-think data alone across all benchmarks, whereas increasing the number of think samples consistently improves performance. This suggests that think data are particularly valuable and highlights the potential benefits of generating and leveraging more think-style supervision.

## F. EP-Order with LLaVA-LLaMA2-7B

We further evaluate EP-Order by replacing the Vicuna-7B decoder in LLaVA-1.5 with LLaMA2-7B, while keeping all other training settings unchanged. The full fine-tuning results are reported in Table 11. We observe several similar trends: (1) the Random paradigm still provides the strongest generalization on average across all benchmarks over other baselines; (2) EP-Order achieves the best performance on most benchmarks and consequently attains the highest average score. Under this backbone, PDPC no longer matches or surpasses the Random paradigm, whereas the Gradient-based ordering attains performance comparable to the Random-based ordering. This further underscores the robustness of Random as a baseline.

## G. Sizes of Mixed Sets and Buffer Samples

The sizes of the mixed sets and buffer samples involve a trade-off between transition smoothness and data purity, since both are extracted from cluster data. We studied six combinations of mixed-set sizes $|c_i|$ and buffer sizes $|b_i|$. Results are summarized in Table 12. In the first setting (top part of the table), we fixed $|b_i|$ and increased $|c_i|$ from 512 to 2048. The best performance is obtained with $|c_i| = $

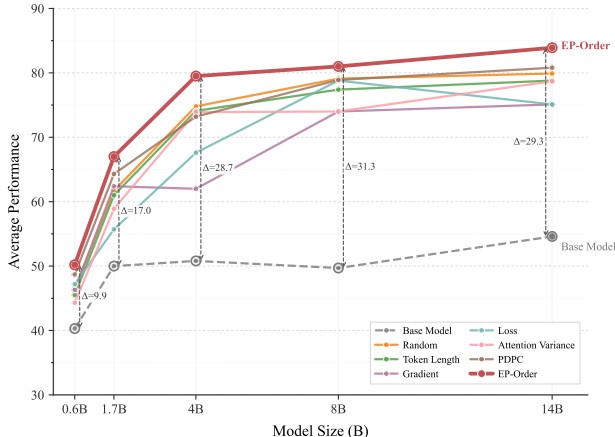

*Figure 7.* Performance visualization of Qwen3-0.6B, Qwen3-1.7B, Qwen3-4B, Qwen3-8B, and Qwen3-14B with different data ordering paradigms.

1024. Next (bottom part of the table), we fixed $|c_i| = 1024$ and varied $|b_i|$ from 128 to 512. The best performance is achieved with $|b_i| = 256$. Consequently, we set $|c_i| = 1024$ and $|b_i| = 256$ in all main experiments. Notably, there is no one-size-fits-all choice for both hyperparameters, as the optimal sizes are related to the total training number and the number of clusters. Empirically, the mixed-set size is more sensitive to data heterogeneity (how abrupt the distribution shift is across clusters), whereas the buffer size primarily governs knowledge retention (how strongly cluster-specific information is preserved).

## H. Clustering with Different Embeddings

By default, we formed clusters using embeddings from the last decoder layer (the 32-nd layer). We varied the layer index $\theta \in \{1, 7, 10, 16, 32\}$ and report results in Table 13. We observed a trend: embeddings from deeper decoder layers produce better clustering and higher performance. For LLaVA-Vicuna-7B, the average score improves from 64.42 to 66.30 when moving from $\theta = 1$ to $\theta = 32$. Although shallower layers are cheaper to compute, we used the last decoder layer for clustering to maximize performance.

## I. Performance Visualization of Model Scaling

We conduct additional experiments with Qwen3-0.6B, 1.7B, 8B, and 14B, together with our previously reported Qwen3-4B results. All models are trained on the Tulu-3-SFT dataset (Lambert et al., 2025). We reported the average performance on ARC, BBH, GSM, and MMLU in the model-scaling figure, and made three observations. (1) Final SFT performance generally improves as the pretrained model scale increases. This trend is particularly clear for EP-Order: its average performance increases from 50.2 to 67.0, 79.5,

81.0, and 83.9 as the base model scales from 0.6B to 1.7B, 4B, 8B, and 14B, respectively. (2) The quality of the pretrained initialization alone is insufficient to explain the final SFT outcome. The average performance of the base model is not strictly monotonic across scales; for example, it remains roughly flat from 1.7B to 8B, while post-SFT performance continues to improve substantially. In particular, the gain of EP-Order over the base model increases from +9.9 at 0.6B to +17.0 at 1.7B, +28.7 at 4B, and +31.3 at 8B. This suggests that larger pretrained models have stronger adaptation capacity during SFT, even when their initialization performance does not improve monotonically. (3) EP-Order performs best across scales. For all evaluated model sizes from 0.6B to 14B, EP-Order achieves the highest average performance among the compared ordering paradigms.

Overall, these results suggest that pretrained model scale substantially affects post-SFT performance, but the raw performance of the base model alone is not a reliable predictor of the final SFT outcome. Larger pretrained models generally benefit more from SFT, and EP-Order remains the most robust and effective data-ordering paradigm across scales.

## J. Impact of EP-Order on Stochastic Optimization Theories

Let $F(\theta) = \frac{1}{n} \sum_{i=1}^{n} f_i(\theta)$ be the full empirical objective, and let $F_{G_t}(\theta) = \frac{1}{|G_t|} \sum_{i \in G_t} f_i(\theta)$ denote the objective of the current group $G_t$ (a cluster or a mixed region).

### J.1. Theories that remain relevant at the local level.

(1) Within a fixed group, without-replacement / random-reshuffling remains the right local lens, since EP-Order still randomly shuffles samples within each cluster and mixed region. A representative example is finite-sum analysis under a random permutation or without-replacement traversal within one epoch (Gürbüzbalaban et al., 2021). Accordingly, the mini-batch gradient is more naturally compared to $\nabla F_{G_t}(\theta_t)$ than to $\nabla F(\theta_t)$; under the usual idealization of uniform sampling from $G_t$, $\mathbb{E}[g_t \mid \theta_t, G_t] = \nabla F_{G_t}(\theta_t)$.

(2) Standard descent-style analyses for the currently optimized objective also remain meaningful, since EP-Order does not modify the optimizer, loss, or standard IT pipeline; it only changes the sampling structure (Bottou et al., 2018).

### J.2. Theories that do not apply directly.

(1) Classical SGD / stochastic first-order analyses based on full-objective unbiasedness (typically under independent/global sampling) do not apply verbatim. A representative example is the standard non-convex SGD framework assuming $\mathbb{E}[g_t \mid \theta_t] = \nabla F(\theta_t)$, where each mini-batch is a globally representative mixture from the whole

dataset (Ghadimi & Lan, 2013). Under EP-Order, this is no longer exact because each mini-batch comes from the current group.

(2) Full-dataset random-reshuffling theory is no longer exact, since EP-Order breaks the assumption that each epoch is a uniform random permutation of all training samples (Safran & Shamir, 2020). Consequently, the standard full-objective noise model $g_t = \nabla F(\theta_t) + \xi_t$, $\quad \mathbb{E}[\xi_t \mid \theta_t] = 0$ is no longer the right description at the level of the full empirical objective.

