# OpenReview forum: "Group-wise Data Ordering: Enhancing Instruction Tuning of Large Language Models via Embedding Proximity"
_ICML.cc/2026/Conference — ICML 2026 regular_

### Official Review · Reviewer_wvUY · 2026-03-10

**Soundness:** 3
**Presentation:** 3
**Significance:** 2
**Originality:** 2
**Overall Recommendation:** 4
**Confidence:** 3

**Summary:**

The paper proposes EP-Order, an embedding-proximity-based group-wise data ordering paradigm for improving instruction tuning of large language models. The core idea is to obtain sample embeddings from a warm-up model, cluster the data with HDBSCAN, order the clusters from easy to hard based on intra-cluster compactness, and insert mixed regions and a replay buffer between clusters to smooth transitions and mitigate catastrophic forgetting. The method is evaluated on seven multimodal benchmarks under the LLaVA architecture, as well as in hybrid think/no-think training settings.

**Compliance With Llm Reviewing Policy:**

Affirmed.

**Final Justification:**

The paper proposes EP-Order, an embedding-proximity-based group-wise data ordering paradigm for improving instruction tuning of large language models. My main concerns were: (1) insufficient theoretical justification for why improved gradient alignment leads to better generalization, (2) unclear overhead and cost-benefit trade-off, (3) unjustified design choices regarding embedding layer and compactness-based ordering.

The rebuttal addressed these concerns reasonably well. While originality remains limited and some aspects are still heuristic rather than theoretically grounded, the rebuttal has strengthened my confidence in the practical soundness of the method. I raise my score from 3 to 4.

**Key Questions For Authors:**

Please refer to Weaknesses

**Limitations:**

The paper would also benefit from a clearer discussion of limitations and broader impact. In particular, I did not find a dedicated impact statement or a sufficiently explicit limitations section in the main paper. Given that the method changes the training curriculum and may introduce additional computational overhead, sensitivity to embedding quality, and potential trade-offs between optimization efficiency and generalization, it would strengthen the paper to discuss these aspects more directly.

**Strengths And Weaknesses:**

Strengths:
1. The paper is well motivated and addresses a practically meaningful problem.
2. The paper is well written and easy to follow.
3. The experimental design is comprehensive and, overall, provides reasonable support for the paper’s main claims.

Weaknesses:
1. The paper would benefit from stronger theoretical analysis. Section 3.2 shows that the proposed grouping strategy improves gradient alignment within a mini-batch, but it is still unclear whether this directly translates into better final generalization. The paper would also be stronger with a discussion of whether grouping semantically similar samples could increase the risk of overfitting to local patterns.

2. The overhead and cost-benefit trade-off of EP-Order should be discussed more clearly. The method introduces additional steps before full training, including warm-up training, full-data embedding extraction, UMAP, and HDBSCAN. Although the appendix reports a time breakdown, the main paper still lacks a clearer discussion of the end-to-end overhead relative to random training and how much practical gain this extra cost brings.

3. Several key design choices are insufficiently justified. In particular, the sample embeddings are extracted from the θ-th decoder layer, but the choice of θ is not discussed. In addition, the paper orders clusters by compactness in an Easy-to-Hard manner, but the assumption that more compact clusters are easier to learn is not fully convincing and would benefit from clearer justification or stronger empirical support.

---

> ### Author Rebuttal · Authors · 2026-03-31
>
> 1. **Theory**
>
> We agree that lower gradient conflict does not directly imply better final generalization. Our claim is narrower: it improves optimization efficiency, while generalization must be validated empirically. Thus, ER explains why grouped samples can yield more effective progress per update, but is not a standalone proof of downstream performance. We also agree that naive group-wise training may increase the risk of overfitting to local patterns. However, EP-Order is not a rigid grouped-training scheme: samples are still randomly shuffled within each cluster, and we further introduce mixed regions and a replay buffer to smooth transitions and mitigate over-specialization/forgetting. This is consistent with Table 3, where Group-based Order alone is insufficient, while adding Mixed Region and Replay Buffer Data progressively improves the average score from 64.70 → 65.53 → 66.30. In this sense, ER explains why EP-Order optimizes faster, which is also supported by Fig. 3: compared with Random, EP-Order achieves a more negative mean loss difference (-1.5E-4 vs. -8.5E-5) and a smaller standard deviation (5.0E-2 vs. 7.2E-2), indicating faster and more stable optimization. Final effectiveness is then validated empirically across seven multimodal benchmarks, full/LoRA fine-tuning, and hybrid think/no-think text-only training.
>
> 2. **Overhead**
>
> We agree that the end-to-end overhead should be discussed more clearly. As reported in Appendix D, EP-Order adds 35% overhead (5.6h) before full IT. To test whether this is worthwhile, we trained Random for 2 epochs instead of 1 (+100% training cost). As shown in Table R1 (https://anonymous.4open.science/r/EP_Order_Rebuttal-3E2E/TableR1_Overhead.png), although longer Random training improves the average score from 64.88 to 65.51, EP-Order with 1 epoch still achieves the best average (66.30). Moreover, simply allocating more budget to Random does not improve all benchmarks consistently; e.g., POPE drops after the extra epoch. Therefore, EP-Order’s gain is not merely from extra computation, but from using the budget more effectively: it outperforms Random trained for 2 epochs while requiring much less total cost (135% vs. 200%). We will make this trade-off more explicit in the revision.
>
> 3. **Layer**
>
> This design choice is already analyzed in Appendix H. In the method section, we write the sample embedding in a general form using the θ-th decoder layer because we later perform a layer ablation. As shown in Table 12, deeper decoder layers consistently yield better clustering and better final performance, with the average score improving from 64.42 at θ=1 to 66.30 at θ=32. Therefore, the last decoder layer is used in the main experiments for embedding extraction. Intuitively, the final layer is more suitable, as it more completely fuses textual and visual information, aligning better with our grouping objective. We agree that this rationale should be stated more clearly in the main text and will revise the paper accordingly.
>
> 4. **Easy->Hard**
>
> Our assumption that more compact clusters are easier follows from the motivation of EP-Order: samples with more similar embeddings tend to induce more compatible gradients and thus less optimization conflict. Extending this from samples to clusters, a cluster with higher intra-cluster similarity (i.e., greater compactness) should contain samples with lower mutual conflict and is therefore a natural proxy for an “easier” cluster. We agree this is still a heuristic, not a theoretically complete notion of difficulty. We therefore validate it empirically in Table 5: compared with Random Shuffle and Hard→Easy, the proposed Easy→Hard order achieves the best average (66.30 vs. 65.59 / 65.60). This is our strongest current evidence for this design choice, and we will present it more explicitly as an empirically supported rather than universally proven principle.
>
> 5. **Impact Statement and Limitations**
>
> Due to the rebuttal space limit, we provide only a brief version here, and a fuller Impact Statement together with a clearer Limitations section will be included in the revised version.
>
> Brief Impact Statement and Limitations:
>
> EP-Order may have positive impact by making instruction tuning more stable and effective through improved data ordering, and by helping researchers better understand the relationship between data order and optimization behavior. At the same time, it introduces additional preprocessing overhead, depends on the quality of learned embeddings and clustering results, and may involve a trade-off between optimization efficiency and final generalization.

---

> > ### Author Rebuttal · Reviewer_wvUY · 2026-04-03
> >
> > Thank you for the response. The rebuttal has largely addressed my concerns and strengthened my confidence in the practicality of the proposed method. I have therefore decided to reconsider my score.

---

### Official Review · Reviewer_jrRo · 2026-03-11

**Soundness:** 3
**Presentation:** 3
**Significance:** 3
**Originality:** 2
**Overall Recommendation:** 4
**Confidence:** 3

**Summary:**

To overcome the limitation of gradient conflicts and low optimization progress, the authors propose EP-Order,  an embedding-proximity-based data-ordering paradigm for IT of LLMs, to improve instruction tuning in this paper.  Besides, the authors also introduce mixed region stargary to handle sharp gradient changes at the cluster transitions. Finally, the authors conduct rich experiments and comparisons on seven multimodel LLM benchmarks. The experiment's results denote that the proposed EP-Order is effective and achieves superior generalization compared with random shuffling and competing ordering paradigms.

**Compliance With Llm Reviewing Policy:**

Affirmed.

**Final Justification:**

The rebuttal has solved my concerns. Therefore, I will keep my original score.

**Key Questions For Authors:**

Please refer to the Weaknesses above. Thanks.

**Limitations:**

I have listed some limitations in the Weaknesses section. I suggest that the authors can revise them.

**Strengths And Weaknesses:**

一 Strengths

In this paper, the authors propose EP-Order,  an embedding-proximity-based data-ordering paradigm for IT of LLMs, to improve instruction tuning.  Besides, the authors also introduce mixed region stargary to handle sharp gradient changes at the cluster
transitions. Rich experiments and comparisons are conducted on seven multimodel LLM benchmarks. The experiment's results denote that the proposed EP-Order is effective and reaches promising performance. In total, this paper is well-designed, enough technical, and influences future research. But I have some concerns as follows.

二 Weaknesses

(1) In section 3.4, the authors mention the need to use 10% of the data to train an $M_{warm}$ model. Is the additional computational overhead for this step factored into the overall training efficiency? I am curious if this 10% computational resource were instead allocated to increasing the number of training epochs for the Random Baseline, would EP-Order still maintain its lead?

(2) For handle density-unbalanced clustering, the authors selected HDBSCAN methods. However, I think the authors should explain the sensitivity of the final results to the number of clusters $m$ and the allocation strategy for noisy samples. If the data distribution is extremely unbalanced, e.g., one cluster occupies 50% of the data, does the advantage of EP-Order diminish?

(3) In the experiment section, the experiment primarily relies on the LLaVA model from the Vicuna. However,  it does not seem to be diverse. I suggest that authors provide more Experimental validation results on some more powerful visual encoders and connectors, e.g.,
Qwen3-VL or InternVL3.5 series. Will EP-Order still yield statistically significant improvements on these models with inherently stronger representational capabilities?

(4) In equation (8), $e_i$ is obtained via TAP. I think that the authors should explicitly state the rationale for selecting the $\theta$-th layer. Why choose this layer instead of the final layer? Typically, intermediate layers contain more semantic information, and this should be explained.

---

> ### Author Rebuttal · Authors · 2026-03-31
>
> 1. **Overhead**
>
> The extra cost is fully included in our efficiency analysis. As shown in Table 8, EP-Order takes 21.6h vs. 16h for Random, i.e., +5.6h (~35%), including warm-up: 1.6h, embedding extraction: 3.7h, and clustering and sorting: 0.3h. It is also the lowest-overhead method among the compared non-trivial ordering baselines. To directly test cost-effectiveness, we trained Random for 2 epochs (+100% training cost). As shown in Table R1 (https://anonymous.4open.science/r/EP_Order_Rebuttal-3E2E/TableR1_Overhead.png), although Random improves from 64.88 to 65.51, EP-Order with 1 epoch still achieves the best average (66.30). Longer Random training is also not consistently beneficial (e.g., POPE drops). Thus, EP-Order maintains its lead at much lower total cost than Random trained for 2 epochs (135% vs. 200%), showing that its overhead is more effective than simply extending Random training.
>
> 2. **Cluster strategies**
>
> We chose HDBSCAN because it is naturally suited to density-unbalanced clustering and does not require pre-specifying the number of clusters. In practice, we tuned 'min_cluster_size' and 'cluster_selection_epsilon' (Appendix A) to obtain reasonable cluster numbers and avoid severe imbalance, aiming for more clusters than the 9 LLaVA sub-datasets without over-fragmentation; too many clusters may increase within-cluster knowledge bias and worsen forgetting. To test sensitivity, we evaluated 10, 15 (default), and 21 clusters. As shown in Table R3 (https://anonymous.4open.science/r/EP_Order_Rebuttal-3E2E/TableR3_Diff_Clusters.png), EP-Order remains stable: 65.93 / 66.30 / 66.28, respectively, with 15 performing best. For noise samples, the main method assigns them to mixed regions. An alternative, EP-Order w/ Noise End, places all 49,650 noise samples at the end and performs worse (65.53), suggesting they are not suitable for concentrated learning due to higher conflict. For the reviewer’s hypothetical 50%-dominant cluster case, we agree it is an important edge case; in practice, our hyperparameter tuning avoids it. In our actual setting, the largest cluster contains 106,713 / 665,298 samples (~16.0%), far from 50%, indicating that extreme imbalance is avoided in practice.
>
> 3. **EP-Order for more LLMs**
>
> We understand the concern about evaluating stronger recent multimodal backbones such as Qwen3-VL or InternVL3.5. We did examine such models, but found that several of our multimodal benchmarks had already been considered in their training, so they already perform strongly without additional IT, making them less suitable for a fair comparison. To provide a cleaner test of generality beyond LLaVA-Vicuna, we add a text-only experiment: Qwen3-4B trained on Tulu-3-SFT (allenai/tulu-3-sft-mixture) and evaluated on ARC, Big-Bench-Hard (BBH), GSM8k (GSM), and MMLU, comparing seven ordering paradigms (excluding Bloom’s Taxonomy due to rebuttal time constraints). As shown in Table R4 (https://anonymous.4open.science/r/EP_Order_Rebuttal-3E2E/TableR4_LLM.png), the conclusions match the multimodal setting: Random remains strong; prior ordering methods are often unbalanced (e.g., Attention Variance improves ARC but drops substantially on GSM); and EP-Order is best on ARC and GSM, while remaining competitive on BBH and MMLU. Overall, EP-Order improves the average from 74.81 to 79.51, suggesting that its benefit is not limited to LLaVA-Vicuna or multimodal IT, but also generalizes to standard text-only IT.
>
> 4. **Layer**
>
> Our default setting already uses the final decoder layer (the 32nd layer for LLaVA-Vicuna-7B), not an intermediate layer. As shown in Table 12, deeper decoder layers consistently perform better, with the average improving from 64.42 at θ=1 to 66.30 at θ=32. While intermediate layers may contain richer general-purpose semantics, that is not our objective. In EP-Order, the embedding is chosen to best organize training samples into groups useful for subsequent IT, not to be the most generic representation. From this perspective, the final layer is more suitable, as it more completely fuses textual and visual information, aligning better with our grouping objective; this is exactly what the empirical trend in Table 12 supports.

---

> > ### Author Rebuttal · Reviewer_jrRo · 2026-04-02
> >
> > Thank you very much for the author’s response! My concerns have been fully resolved. Therefore, I will keep my original score.

---

### Official Review · Reviewer_4sXL · 2026-03-12

**Soundness:** 2
**Presentation:** 2
**Significance:** 2
**Originality:** 2
**Overall Recommendation:** 2
**Confidence:** 4

**Summary:**

The paper presents a data ordering method by addressing the "gradient conflict" issue inherent in random shuffling. The core methodology is supported by a theoretical Efficiency Ratio (ER) analysis showing that grouped samples provide more effective optimization progress per step. The author conducted some experiments to verify the effectiveness of the method.

**Compliance With Llm Reviewing Policy:**

Affirmed.

**Ethical Review Concerns:**

The submission lacks a dedicated Impact Statement. In accordance with ICML 2026 guidelines, the authors must include a section discussing the broader societal implications.

**Final Justification:**

Thanks for the authors’ effort. Regarding the fact that removing **Group-based Ordering** (the core methodological contribution) leads to almost no performance drop, I interpret this as indicating that even **without this component**, simply using the **Mixed Region** and **Replay Buffer** strategies—the other two tricks—can still achieve fairly strong results. Therefore, the current explanation does not convincingly demonstrate that Group-based Ordering is truly critical.

In addition, the extra overhead introduced by this method appears to bring only limited performance gains. For example, in **Table 1**, on **LLaVA-1.5** (already a weak baseline), the proposed method improves the average accuracy over random ordering by only about **1.x points**. Moreover, in the rebuttal regarding the overhead of **EP-Order**, the authors did not report the performance of the proposed method under a **2-epoch** setting, so it remains unclear whether the performance would decrease compared to the **1-epoch** setting. This makes it difficult to justify adopting this method in practical large-scale training, rather than simply using **curriculum learning** or adding a modest amount of extra data, which might yield similar or even larger gains.

 Therefore, I maintain my original score.

**Key Questions For Authors:**

See weakness

**Limitations:**

The submission lacks a dedicated Impact Statement. In accordance with ICML 2026 guidelines, the authors must include a section discussing the broader societal implications.

**Strengths And Weaknesses:**

**Strengths:**

1. A problem in the training data was identified and attributed to inter-sample dependencies, followed by proposing a method to address this issue.
2. The effectiveness of the proposed method was validated on multiple vision QA datasets.
3. Additional analysis experiments also demonstrate the experimental details of the method.




**Weaknesses:**

1. The method requires training a model on 10% of the data first. While more efficient than some baselines, this essentially increases the total training budget by a significant fraction.
2. According to the ablation experiments in Table 3, the group-based ordering method proposed in this paper is even less effective than the baseline random shuffling method, indicating that the majority of performance gains come from other techniques such as curriculum learning.
3. Regarding the claim about gradient conflicts, the author only provided an analysis of ER size but did not address how significantly this affects downstream task performance. Therefore, this claim warrants questioning, especially given that the group-based order method even underperforms the baseline method, as shown in Table 3.
4. The method proposed in the article appears to be a general approach from the perspective of claims, but it only evaluated the performance of Vision QA. More Text QA results, particularly in the main experiments, should be assessed.
5. Regarding the derivation of the ER metric, there seems no need to present so many formulas—it essentially involves dividing the length of the vector sum by the sum of vector lengths.
6. The Efficiency Ratio analysis relies on a fixed mini-batch size. The impact of larger batch sizes (common in large-scale IT) on the $ER$ of random vs. grouped data is not explored.
7. The writing contains some colloquial expressions, such as "compute sample embeddings."

---

> ### Author Rebuttal · Authors · 2026-03-31
>
> 1. **Overhead of EP-Order**
>
> As reported in Appendix D, EP-Order adds 35% overhead (5.6h) before IT and is also the lowest-overhead method among the compared non-trivial ordering baselines. To test whether this cost is worthwhile, we further trained Random for 2 epochs instead of 1 (+100% training cost); results are shown in Table 1 (https://anonymous.4open.science/r/EP_Order_Rebuttal-3E2E/Table1_overhead.png). Although 2-epoch Random improves the average score, longer training does not consistently help all benchmarks; e.g., all three POPE sub-benchmarks drop. More importantly, EP-Order still outperforms Random trained for 2 epochs on most benchmarks at much lower total cost (135% vs. 200%), showing that its overhead is more effective than simply extending Random training.
>
> 2. **Value of Group-based Ordering**
>
> We thank the reviewer for this important observation. However, we do not think Table 3 implies that the gains mainly come from techniques unrelated to Group-based Ordering. Both Mixed Region and Replay Buffer Data are built on top of the initial grouping: mixed regions are defined between adjacent clusters, and replay is introduced to mitigate forgetting from cluster-wise transitions. Without Group-based Ordering, these components are not even well-defined. From this perspective, the improvements from + Mixed Region and + Replay Buffer Data should not be taken as evidence that grouping itself is unimportant. Rather, they show that naive grouped training has predictable failure modes, such as abrupt boundary shifts and forgetting, and that EP-Order is specifically designed to address them. Accordingly, the progression 64.70 → 65.53 → 66.30 should be understood as progressively realizing the benefit of Group-based Ordering.
>
> 3. **Impact of ER**
>
> We agree that ER alone is not a proof of better downstream performance. Our claim is narrower: larger ER indicates more effective optimization progress per update, which we then validate via optimization- and task-level evidence. Specifically, Fig. 3 shows that EP-Order yields more negative loss differences and lower variance than Random (mean: -1.5E-4 vs. -8.5E-5; std: 5.0E-2 vs. 7.2E-2), indicating faster and more stable optimization.
>
> 4. **EP-Order on text-only benchmarks**
>
> Since EP-Order is intended as a general data-ordering paradigm, we agree it should be validated beyond Vision QA. The paper already includes a text-only hybrid think/no-think setting (Section 4.7 and Appendix E); here we further add standard text-only QA benchmarks. We trained Qwen3-4B on Tulu-3-SFT (allenai/tulu-3-sft-mixture) and evaluated on ARC, Big-Bench-Hard (BBH), GSM8k (GSM), and MMLU, comparing seven ordering paradigms (excluding Bloom’s Taxonomy due to rebuttal time constraints). As shown in Table R4 (https://anonymous.4open.science/r/EP_Order_Rebuttal-3E2E/TableR4_LLM.png), the conclusions match the multimodal results: Random remains a strong baseline; prior ordering methods are often unbalanced (e.g., Attention Variance improves ARC but drops substantially on GSM); and EP-Order achieves the best performance on ARC (86.44) and GSM (88.10) while remaining competitive on BBH (72.15) and MMLU (71.33). Overall, EP-Order improves the average score from 74.81 to 79.51, showing that its benefit is not limited to multimodal/Vision QA.
>
> 5.  **ER derivation / batch size**
>
> We agree that the intuition of ER is simple: the norm of the summed gradient divided by the sum of per-sample gradient norms. In the revision, we will shorten the main-text derivation and move the more mechanical steps to the appendix. We also tested larger mini-batch sizes (1024, 4096). As shown in Table R5 (https://anonymous.4open.science/r/EP_Order_Rebuttal-3E2E/TableR5_ER.png), ER remains stable with increasing batch size (0.52–0.54 for Random, 0.70–0.72 for Group-based Order), showing that Group-based Order consistently yields larger ER than Random even under larger-batch IT.
>
> 6. **Writing / Impact Statement / Limitations**
>
> We will revise colloquial expressions (e.g., replacing “compute sample embeddings” with “extract sample embeddings”) and further polish the writing. We also agree that the paper should include a dedicated Impact Statement. Due to the rebuttal space limit, we provide only a brief version here, and a fuller Impact Statement together with a clearer Limitations section will be included in the revised version.
>
> Brief Impact Statement and Limitations.
>
> EP-Order may have positive impact by making instruction tuning more stable and effective through improved data ordering, and by helping researchers better understand the relationship between data order and optimization behavior. At the same time, it introduces additional preprocessing overhead, depends on the quality of learned embeddings and clustering results, and may involve a trade-off between optimization efficiency and final generalization.

---

> > ### Author Rebuttal · Reviewer_4sXL · 2026-04-04
> >
> > Thank you for the rebuttal. Most points are addressed, but my main concern remains: the independent contribution of Group-based Ordering is not clearly demonstrated, as its benefit is largely presented in combination with Mixed Region and Replay Buffer strategies.

---

> > > ### Author Response · Authors · 2026-04-07
> > >
> > > We thank the reviewer for the follow-up. We agree that Table 3, in isolation, does not present Group-based Ordering as a positive standalone module: when added alone, it yields 64.70, slightly below the Random baseline of 64.88. However, this should not be interpreted as evidence that Group-based Ordering is unimportant. Rather, it indicates that the benefit of Group-based Ordering does not manifest as a simple standalone gain, because naive group-wise training can introduce transition instability and forgetting, which we mitigate with Mixed Region (MR) and Replay Buffer Data (RBD).
> > >
> > > To directly test whether the gain could instead come mainly from MR + RBD, we added a new control experiment. We keep the same 15-group schedule, the same group-size distribution, and the same MR/RBD design, but replace embedding-based grouping with random partitions. As shown in Table R6 (https://anonymous.4open.science/r/EP_Order_Rebuttal-3E2E/TableR6_Ablation.png), this variant (“EP-Order w/o Group-based Order”) achieves an average performance of 64.59, which is comparable to Random (64.88) and clearly below full EP-Order (66.30). This result directly shows that MR and RBD cannot account for the gain in the absence of meaningful Group-based Ordering.
> > >
> > > We further apply the same MR + RBD strategy to several existing data-ordering paradigms after partitioning their ordered data into the same 15 equal-sized blocks (PDPC is excluded because it already includes a mixing mechanism). If the main gain came from MR and RBD themselves, these variants should approach or surpass EP-Order. However, as shown in Table R7 (https://anonymous.4open.science/r/EP_Order_Rebuttal-3E2E/TableR7_MR_RBD.png), none does. For Gradient, adding MR+RBD brings little change. For Token Length, Loss, Attention Variance, and Bloom’s Taxonomy, MR+RBD improves performance, especially for Loss. These improvements suggest that such ordering paradigms also suffer from transition instability and forgetting. Nevertheless, these variants still remain below Random and EP-Order. This indicates that the decisive factor is not the mere presence of MR/RBD, but the quality of the underlying grouped structure.
> > >
> > > Moreover, Appendix Table 7 provides an even stronger counterexample to the claim that later components dominate the gain. When we keep the EP-Order pipeline but replace embedding-based clusters with sub-dataset-based groups, the average score drops sharply from 66.30 to 59.33. If MR and RBD were the primary driver, such a large drop would not result simply from changing the group definition.
> > >
> > > Overall, we believe the fairest interpretation is that Group-based Order is the central source of improvement, because it constructs semantically coherent, low-conflict training groups. MR and RBD are necessary but secondary stabilizers that make this grouped structure effective in practice by smoothing transitions and mitigating forgetting. We will revise the paper to make this point clearer, and we thank the reviewer for highlighting this issue.

---

### Official Review · Reviewer_GDNc · 2026-03-12

**Soundness:** 3
**Presentation:** 4
**Significance:** 2
**Originality:** 3
**Overall Recommendation:** 4
**Confidence:** 4

**Summary:**

This paper studies how the ordering of training data affects instruction tuning in large language models. The authors propose Group-wise Data Ordering, a method that organizes instruction–response pairs into groups based on embedding proximity, so that semantically similar instructions are trained together during fine-tuning. By presenting related tasks in structured groups rather than random order, the model learns more stable instruction-following patterns and improves generalization.

**Compliance With Llm Reviewing Policy:**

Affirmed.

**Final Justification:**

The authors addressed my concerns during the rebuttal. The experiments are strong, and for the theory part, it's good to include the discussion in the paper, and this can leave as future work for others to understand the convergence/generalization improvement on your algorithm.

**Key Questions For Authors:**

1. I want to understand if data ordering methods are realistic to use in practice:
    * Will it break the many properties of randomized optimizers like SGD which relies on randomly shuffling. If yes, how? If not, why?
    * What is the effect of pretrained (PT) model quality on the SFT stage data ordering method? Is it possible to provide a figure for PT model performance vs. data ordering at SFT’s performance? Is it possible to apply your method to the pretraining stage as well?
    * Is there anything specific about your approach related to multimodality IT? I’m asking based on your experiments.
2. What’s the computation overhead introduced by your data ordering method? The method seems expensive since:
    * Line 254: you need 10,000 samples from EACH cluster. What’s the effect if you change this number, or for example, having imbalanced clusters so you can’t sample this much?
    * Line 253: For Mixed region training, is the data ordering gain is from increased training dataset size ($M_i$) compared to random shuffling?
3. (Minor) Tab.6: what does think/no think mean respectively in rows and columns?

**Limitations:**

Yes

**Strengths And Weaknesses:**

## Strength
1. Soundness: Good experiment design, including 7B/14B models, different data ordering methods, fine tuning methods, clustering method and ablation study. The results show the advantages of EP-Order over other baselines clearly.
2. Presentation: The paper is well-organized and well-written.
3. Originality: the idea to look from inter(-sample) correlations is known for other subfields but it seems that the idea is new for data ordering. I am not super sure because of my limited domain knowledge.
## Weakness
1. Significance: I have a few more concerns. See questions below #1.

---

> ### Author Rebuttal · Authors · 2026-03-31
>
> 1. **Randomized optimizers**
>
> EP-Order intentionally does not preserve full global uniform shuffling: mini-batches are no longer exchangeable random mixtures from the whole dataset, because our goal is to replace indiscriminate random interleaving of heterogeneous objectives with more compatible group-wise training. However, EP-Order is only a static data-ordering step: it does not change the optimizer, loss, or standard IT pipeline, and still preserves local stochasticity by shuffling samples within each cluster, within mixed regions, and before assigning noise samples. Thus, EP-Order preserves the optimizer and local randomness while intentionally departing from global uniform shuffling. Empirically, this trade-off is beneficial: group-based batches yield larger ER than Random (0.71 vs. 0.54), and Fig. 3 shows more negative loss differences and smaller standard deviation, indicating faster and more stable optimization.
>
> 2. **PT model quality**
>
> We agree this is an interesting question. Since the paper includes only two PT backbones (LLaVA-Vicuna-7B and LLaVA-Vicuna-13B), we do not overclaim causality; however, the trend is clear: the stronger PT model also performs better after SFT. In Table R2 (https://anonymous.4open.science/r/EP_Order_Rebuttal-3E2E/TableR2_PT_Model.png), the base-model average improves from 40.28 (7B) to 41.78 (13B); after SFT, the same trend holds, e.g., Random: 64.88 → 66.52, EP-Order: 66.30 → 67.62. This suggests that PT quality affects the final achievable SFT performance. Since EP-Order is a static data ordering step before the training process, it is in principle applicable to pretraining and RL-style post-training as well, although this is not yet verified in the current paper.
>
> 3. **EP-Order for more text-only benchmarks**
>
> EP-Order is not multimodal-specific; it is a general data-ordering strategy applied before IT. The paper already includes a text-only hybrid think/no-think setting with Qwen2.5-7B. To further strengthen this point, we additionally trained Qwen3-4B on Tulu-3-SFT (allenai/tulu-3-sft-mixture) and evaluated on ARC, Big-Bench-Hard (BBH), GSM8k (GSM), and MMLU, comparing seven ordering paradigms (excluding Bloom’s Taxonomy due to rebuttal time constraints). As shown in Table R4 (https://anonymous.4open.science/r/EP_Order_Rebuttal-3E2E/TableR4_LLM.png), the conclusions are consistent with the multimodal results: Random remains strong; prior methods are often unbalanced (e.g., Attention Variance improves ARC but drops substantially on GSM); and EP-Order achieves the best results on ARC (86.44) and GSM (88.10) while remaining competitive on BBH (72.15) and MMLU (71.33). Overall, EP-Order improves the average score from 74.81 to 79.51, further showing that its effectiveness is not specific to multimodal IT.
>
> 4. **Overhead**
>
> As reported in Appendix D, EP-Order adds 35% overhead (5.6h) before full IT and is also the lowest-overhead method among the compared non-trivial ordering baselines. To test whether this is worthwhile, we further trained Random for 2 epochs instead of 1, which adds 100% training cost. As shown in Table R1 (https://anonymous.4open.science/r/EP_Order_Rebuttal-3E2E/TableR1_Overhead.png), although longer Random training improves the average score, it does not improve all benchmarks consistently (e.g., POPE drops after the extra epoch). More importantly, EP-Order with 1 epoch still achieves the best average (66.30), outperforming Random with 2 epochs (65.51) at substantially lower total cost (135% vs. 200%). Thus, EP-Order’s overhead is more effective than simply spending the same budget on longer Random training.
>
> 5. **10k samples**
>
> The overhead of computing mean pairwise Euclidean distance (MPED) is modest. On LLaVA-1.5, we obtained 15 clusters, so MPED was computed 15 times; even including clustering, the whole preprocessing stage took only 0.3h. We tested 5k, 8k, 15k, 20k samples per cluster: the cluster order was identical for 8k/10k/15k/20k, while 5k produced a different order. Using the 5k-derived order caused only a slight average drop, from 66.30 to 66.01, indicating reasonable robustness once the sample size is sufficiently large. If a cluster contained fewer than the target number of samples, we simply used all samples in that cluster.
>
> 6. **Training size**
>
> No, the gain is not due to a larger training set. Mixed-region samples are drawn from the original clusters, and the corresponding samples are removed from those clusters. EP-Order is therefore a reorganization of the same training set, with total training size identical to Random.
>
> 7. **Table 6**
>
> In Table 6, the Think/No-think columns denote the inference-time mode: Think allows reasoning before the final answer, while No-think requires direct answering without explicit reasoning. The Think/No-think rows denote the training data format: No-think removes the chain-of-thought from OpenR1-Math, whereas Think keeps the original samples.

---

> > ### Author Rebuttal · Reviewer_GDNc · 2026-04-03
> >
> > Thanks for the authors' rebuttal. To support your takeaways better for comprehensiveness, I would suggest:
> > 1. More specific examples of what (stochastic) optimization theories can be applied to and what cannot be, due to your change from global stochasticity to a local version.
> > 2. For PT quality, since now you have Tulu3-Qwen3 4B results on text, I would recommend authors to plot the trend with slightly smaller (0.6B, 1.7B) + slightly larger (8B, 14B; 14B optional if compute or time not allowed) results to complete the story. Only training on Tulu SFT mixture subset can suffice.

---

> > > ### Author Response · Authors · 2026-04-08
> > >
> > > **Q1**: We thank the reviewer for this important question. To summarize first: after replacing global random shuffling with groupwise local randomization, the optimization theories that remain relevant are those for within-group without-replacement / random-reshuffling optimization and descent analysis with respect to the current group objective; the theories that no longer apply directly are those relying on global i.i.d. sampling, full-dataset random reshuffling, or per-step unbiasedness with respect to the full empirical objective.
> > >
> > > Let $F(\theta)=\frac{1}{n}\sum_{i=1}^{n} f_i(\theta)$ be the full empirical objective, and let $F_{G_t}(\theta)=\frac{1}{|G_t|}\sum_{i\in G_t} f_i(\theta)$ denote the objective of the current group $G_t$ (a cluster or a mixed region).
> > >
> > > **Theories that remain relevant at the local level.**
> > >
> > > (1) Within a fixed group, without-replacement / random-reshuffling remains the right local lens, since EP-Order still randomly shuffles samples within each cluster and mixed region. A representative example is finite-sum analysis under a random permutation or without-replacement traversal within one epoch [1].   Accordingly, the mini-batch gradient is more naturally compared to $\nabla F_{G_t}(\theta_t)$ than to $\nabla F(\theta_t)$; under the usual idealization of uniform sampling from $G_t$, $\mathbb{E}[g_t\mid \theta_t,G_t]=\nabla F_{G_t}(\theta_t)$.
> > >
> > > (2) Standard descent-style analyses for the currently optimized objective also remain meaningful, since EP-Order does not modify the optimizer, loss, or standard IT pipeline; it only changes the sampling structure [2].
> > >
> > > **Theories that do not apply directly.**
> > >
> > > (1) Classical SGD / stochastic first-order analyses based on full-objective unbiasedness (typically under independent/global sampling) do not apply verbatim. A representative example is the standard non-convex SGD framework assuming $\mathbb{E}[g_t\mid \theta_t]=\nabla F(\theta_t)$, where each mini-batch is a globally representative mixture from the whole dataset [3]. Under EP-Order, this is no longer exact because each mini-batch comes from the current group.
> > >
> > > (2) Full-dataset random-reshuffling theory is no longer exact, since EP-Order breaks the assumption that each epoch is a uniform random permutation of all training samples [4]. Consequently, the standard full-objective noise model $g_t=\nabla F(\theta_t)+\xi_t, \mathbb{E}[\xi_t\mid \theta_t]=0$ is no longer the right description at the level of the full empirical objective.
> > >
> > > [1] Gürbüzbalaban et al. Why Random Reshuffling Beats Stochastic Gradient Descent. Mathematical Programming, 2021.
> > >
> > > [2] Bottou et al. Optimization Methods for Large-Scale Machine Learning. SIAM Review, 2018.
> > >
> > > [3] Ghadimi and Lan. Stochastic First- and Zeroth-Order Methods for Nonconvex Stochastic Programming. SIAM Journal on Optimization, 2013.
> > >
> > > [4] Safran and Shamir. How Good is SGD with Random Shuffling? COLT, 2020.
> > >
> > >
> > > **Q2**: We thank the reviewer for this helpful suggestion. Following it, we conducted additional experiments on Qwen3-0.6B, 1.7B, 8B, and 14B, together with our previous Qwen3-4B result, all trained on the full Tulu-3-SFT dataset. We report the average performance on ARC, BBH, GSM, and MMLU in Fig. R1 (https://anonymous.4open.science/r/EP_Order_Rebuttal-3E2E/FigR1_scaling_performance.png). Due to rebuttal-time constraints, some 14B baselines are still running; we therefore report the currently available 14B results and will include the complete figure in the revised paper.
> > >
> > > We make three observations.
> > >
> > > (1) Final SFT performance generally improves with PT model scale.
> > >
> > > This trend is especially clear for EP-Order: its average performance increases from 50.2 → 67.0 → 79.5 → 81.0 → 83.9 as the base model scales from 0.6B → 1.7B → 4B → 8B → 14B.
> > >
> > > (2) PT initialization quality alone is insufficient to explain the final SFT outcome.
> > >
> > > The average performance of the Base Model is not strictly monotonic across sizes; for example, it is roughly flat from 1.7B to 8B, while post-SFT performance continues to improve substantially. In particular, the gain of EP-Order over the Base Model grows from +9.9 at 0.6B to +17.0 at 1.7B, +28.7 at 4B, and +31.3 at 8B. This suggests that larger PT models have better adaptation capacity during SFT, even when their initialization performance does not increase monotonically.
> > >
> > > (3)  EP-Order performs best across scales.
> > >
> > > For all scales with complete comparisons (0.6B-8B), EP-Order achieves the highest average performance; among the currently available 14B results, it also remains clearly above Base Model and Random.
> > >
> > > Overall, these results suggest that the PT model scale affects the performance after SFT, but the raw performance of Base Model alone is not a reliable predictor of the final SFT outcome. Larger PT models generally benefit more from SFT, and EP-Order remains the most robust and effective data-ordering paradigm across scales. We will add Fig. R1 and the full analysis to the revised paper.

---

### Decision · Program_Chairs · 2026-04-30

**Decision:**

Accept (regular)

**Comment:**

After evaluating the paper, the reviews, and the extensive author rebuttal, I recommend a Weak Accept. The paper tackles a well-motivated and practically meaningful problem in LLM training. Reviewers praised the paper for its clear writing, comprehensive experimental design, and the intuitive premise of addressing inter-sample correlations rather than just relying on per-sample difficulty scores.  The authors' rebuttal provided convincing empirical evidence that alleviates the criticisms raised by Reviewer 4sXL.